# Provably Data-driven Multiple Hyper-parameter Tuning with Structured Loss Function

**Tung Quoc Le** [1]   **Anh Tuan Nguyen** [2]   **Viet Anh Nguyen** [3]

## Abstract

Data-driven algorithm design automates hyperparameter tuning, but its statistical foundations remain limited because model performance can depend on hyperparameters in implicit and highly non-smooth ways. Existing guarantees focus on the simple case of a one-dimensional (scalar) hyperparameter. This leaves the practically important, multi-dimensional hyperparameter tuning setting unresolved. We address this open question by establishing the first general framework for establishing generalization guarantees for tuning multi-dimensional hyperparameters in data-driven settings. Our approach strengthens the generalization guarantee framework for semi-algebraic function classes by exploiting tools from real algebraic geometry, yielding sharper, more broadly applicable guarantees. For completeness, we also instantiate the first lower bound for this general setting. We further extend the analysis to hyperparameter tuning using the validation loss under minimal assumptions, and derive improved bounds when additional structure is available. Finally, we demonstrate the scope of the framework with new learnability results, including data-driven weighted group lasso and weighted fused lasso.

## 1. Introduction

Hyperparameter tuning (a Ilemobayo et al., 2024) is a cornerstone of the modern machine learning paradigm, often determining a model's performance. Despite its critical role, this tuning process is often regarded more as an art than a rigorous scientific endeavor. In practice, the vast majority of practitioners rely on naive grid search, which discretizes

[1]Université Grenoble Alpes, LJK, CNRS, Grenoble INP, 38000 Grenoble, France [2]Carnegie Mellon University, Machine Learning Department [3]Chinese University of Hong Kong, Department of Systems Engineering and Engineering Management. Correspondence to: Anh Tuan Nguyen <atnguyen@cs.cmu.edu>.

*Proceedings of the 43rd International Conference on Machine Learning*, Seoul, South Korea. PMLR 306, 2026. Copyright 2026 by the author(s).

continuous hyperparameter spaces and selects the candidate that yields the best empirical results. While straightforward, this exhaustive approach is theoretically unprincipled and offers no performance guarantees.

To automate this tedious task, recent literature has introduced computational techniques such as Bayesian optimization and random search. While empirically effective, these methods lack robust theoretical foundations or rely on restrictive assumptions. For instance, Bayesian optimization (Bergstra et al., 2011; Snoek et al., 2012) generally assumes that a model's performance can be approximated as a noisy evaluation of a smooth, expensive function. In reality, this assumption could fail due to the volatile loss landscapes of complex models. Furthermore, these methods introduce their own complexity, requiring the selection of meta-hyperparameters, such as the acquisition function, kernel type, and bandwidth. Similarly, other approaches, including random search and spectral methods (Bergstra & Bengio, 2012; Hazan et al., 2018), often provide theoretical guarantees only for discrete, finite grids. Meanwhile, advanced resource-allocation strategies like Hyperband (Li et al., 2018), which are effective at accelerating search via early stopping, primarily optimize computational efficiency for a fixed problem instance over discrete search spaces. Consequently, they do not characterize the fundamental theoretical complexity of identifying optimal continuous parameters. This gap highlights the pressing need for a rigorous theoretical foundation for hyperparameter tuning.

To address this gap, the *data-driven algorithm design* framework models the hyperparameter tuning problem as a statistical learning problem over an unknown, application-specific distribution $\mathcal{D}$ of problem instances (Balcan, 2020; Gupta & Roughgarden, 2020). From a finite set of training instances drawn from $\mathcal{D}$, we seek to identify the hyperparameter configuration that is provably guaranteed to perform well on future, *un*seen instances drawn from the same distribution. To this end, we define the loss function $\ell_\alpha(x)$ via a bi-level optimization structure:

$$\ell_\alpha(x) = \inf_{\theta \in \mathcal{S}(x,\alpha)} g(x, \alpha, \theta),$$
$$\text{where } \mathcal{S}(x,\alpha) \triangleq \arg\min_{\theta \in \Theta} f(x, \alpha, \theta). \tag{1}$$

Here, $f$ represents the *training objective* (i.e, the optimiza-

tion surrogate), while $g$ represents the *validation objective* (i.e., the target metric). For instance, consider the linear regression problem in which we would like to tune the (one-dimensional) ridge regularization parameter. The problem instance can be represented as $x = (A, b, A', b')$ containing both the training set $(A, b)$ and the validation set $(A', b')$, which could be extracted from the available data. The training objective $f(x, \alpha, \theta)$ explicitly includes the regularization term $\|A\theta - b\|_2^2 + \alpha\|\theta\|_2^2$ to stabilize the solution, whereas the validation objective $g(x, \alpha, \theta) = \|A'\theta - b'\|_2^2$ evaluates performance with*out* the penalization. In this context, the regularization parameter $\alpha$ shapes the optimization landscape of the training objective $f$, but affects the validation objective $g$ only *implicitly* through the learned parameters. Analyzing this implicit dependency is the core challenge that we need to overcome.

The approach we take in this paper leverages statistical learning theory to bound the learning-theoretic complexity of the family of loss functions induced by the hyperparameters. Formally, it behooves us to analyze the learning-theoretic complexity (e.g., pseudo-dimension) of the function class $\mathcal{L} = \{\ell_\alpha : \mathcal{X} \to [-H, H] \mid \alpha \in \mathcal{A}\}$, where $\mathcal{A}$ is the hyperparameter space and $H$ is a positive-valued bound of the loss functions. The analysis in Balcan et al. (2025) leverages the observation that for many problems of interest, given the problem instance $x$, $f_x(\alpha, \theta) \triangleq f(x, \alpha, \theta)$ admits a *piecewise polynomial structure* with respect to $\alpha$ and $\theta$ (see Definition 3.1 and Figure 1 for a simple illustration). This structural assumption is ubiquitous: it has been established for a wide range of applications in both classical learning theory (Bartlett et al., 1998; Montúfar et al., 2014; Bartlett et al., 2019) as well as data-driven algorithm design problems (Balcan et al., 2021a; 2023; Nguyen & Nguyen, 2026; Cheng & Basu, 2025).

On the downside, the result from Balcan et al. (2025) suffers from numerous limitations that hinder its general applicability. First, their analysis is strictly confined to one-dimensional hyperparameters (i.e., $\mathcal{A} = \mathbb{R}$) based on ad hoc geometric arguments (e.g., oscillations, monotonic curves). In reality, many ML/AI pipelines rely on stacking multiple regularization terms to achieve the desired effects; the most popular is the elastic net, which combines L1 (lasso) and L2 (ridge) penalties with different hyperparameter weights. However, this geometric approach breaks down in this multi-dimensional setting ($\alpha \in \mathbb{R}^p$), where critical sets become high-dimensional manifolds rather than simple curves. Second, their results are limited to the simple setting where the training and evaluation objectives are identical (i.e., $f \equiv g$). For the ridge regression illustrated above, this requirement imposes that the training and validation sets are identical, i.e., $(A, b) = (A', b')$. This restriction violates the fundamental principles of model selection, which require the training and validation sets to be well separated to prevent

overfitting and produce an unbiased estimate of the performance. Third, their approach requires strong regularity assumptions on the boundaries of the piecewise structure (e.g., ELICQ conditions in Balcan et al. (2025, Assumption 1)). Finally, there is no lower bound in the general setting presented in the prior work, which makes it hard to assess the tightness of the upper bound. These limitations motivate the development of the general model-theoretic framework we present in this paper.

**Contributions.** In this work, we provide an affirmative answer to the open question posed by Balcan et al. (2025) by establishing the concrete sample complexity for the function class $\mathcal{L} = \{\ell_\alpha : \mathcal{X} \to [-H, H] \mid \alpha \in \mathcal{A}\}$, where $\mathcal{A} \subset \mathbb{R}^p$ with $p \geq 1$. Our contributions are as follows:

- We establish in Theorem 4.1 a general tool that connects the learning-theoretic complexity of a function class $\mathcal{L}$ to the logical complexity of its functions $\ell_\alpha$. Specifically, we show that if the loss function can be described by a polynomial *first-order logic*, its pseudo-dimension is bounded by the complexity of the quantifier elimination process (Basu et al., 2006).

- Second, we apply this framework to the training-loss setting (i.e., $f \equiv g$) studied by Balcan et al. (2025). We show in Theorem 5.1 that if the objective $f(x, \alpha, \theta)$ admits a piecewise polynomial structure for any given problem instance $x$, the induced loss $\ell_x(\alpha) = \min_{\theta \in \Theta} f(x, \alpha, \theta)$ is expressible as a polynomial FOL formula. This allows us to derive the first generalization guarantees for multi-dimensional hyperparameters ($\alpha \in \mathbb{R}^p$), effectively resolving the main open question in prior work. Moreover, we also present the first pseudo-dimension lower-bound in Theorem 5.2, showing that our upper-bound is tight in some scenarios.

- Third, we extend our analysis to the general bi-level validation tuning setting ($f \neq g$). We prove in Theorem 6.1 that validation loss functions are definable in FOL, ensuring learnability under minimal assumptions.

- Forth, we show in Section 7 how we can exploit the optimal solution path structure to improve the bounds.

- Finally, we demonstrate in Section 8 the versatility of our framework by establishing the first learnability guarantees for two new applications: data-driven tuning for weighted group LASSO and weighted fused LASSO, some of which go *beyond the piecewise polynomial assumptions*.

**Technical challenges.** The key challenge in our setting is that the loss function $\ell_\alpha(x)$ is defined implicitly via a bi-level optimization problem involving non-convex, non-smooth objectives. Consequently, the dependence of $\ell_\alpha(x)$

can be highly volatile, exhibiting discontinuities and potentially singularities.

To overcome these challenges, we adopt a *model-theoretic* perspective, drawing on tools from real algebraic geometry. Instead of analyzing the local geometry of the solution paths (i.e., $\mathcal{S}(x, \alpha) = \arg\min_{\theta \in \Theta} f(x, \alpha, \theta)$), we view the optimality conditions of the inner problem as a predicate in polynomial first-order logic (FOL). After that, we use *quantifier-elimination* technique, transforming polynomial FOLs to a *quantifier-free formula* (QFF), of which the computation can then be described as a Goldberg-Jerrum (GJ) algorithm (Goldberg & Jerrum, 1993; Bartlett et al., 2022). This allows us to use the GJ framework to establish the generalization guarantee for the function class of interest

**Notations.** We denote $\text{sign}(t) = 0$ if $t = 0$, 1 if $t > 0$, and $-1$ if $t < 0$. For $z \in \mathbb{R}^k$, we denote $\mathbb{R}[z]$ the polynomial ring in $z$ over $\mathbb{R}$, i.e., $P(z) \in \mathbb{R}[z]$ is a (multivariate) polynomial of $z$. We denote $\deg(P)$ the degree of the polynomial $P$. For a multiple inputs functions $h(z_1, z_2)$, given a fixed component, says $z_1$, we denote $h_{z_1}(z_2) \triangleq h(z_1, z_2)$ is an induced function where we treat $z_1$ as fixed and $z_2$ as the only variables.

### 1.1. Related Works

**Data-driven hyperparameter tuning**, initiated by (Balcan, 2020; Gupta & Roughgarden, 2020), frames hyperparameter tuning as a multi-task learning problem. This data-driven perspective has demonstrated significant empirical success across diverse fields, including low-rank approximation (Li et al., 2023; Indyk et al., 2019), accelerated linear system solvers (Sakaue & Oki, 2024; Nguyen & Nguyen, 2026), and branch-and-cut strategies for (mixed integer) linear programming (Balcan et al., 2021b), among others.

**Theoretical analysis for data-driven hyperparameter tuning.** The practical success of data-driven hyperparameter tuning has motivated a line of theoretical work aimed at establishing rigorous generalization guarantees (Balcan et al., 2021a;b; Bartlett et al., 2022; Balcan et al., 2023; 2024). Analyzing these problems is generally more challenging than standard statistical learning theory due to the volatile, non-smooth dependence of the loss $\ell_x(\alpha)$ on the hyperparameters $\alpha$. Most prior guarantees focus on settings where the algorithm's behavior is determined solely by $\alpha$, typically by exploiting the piecewise structure of $\ell_x(\alpha)$ w.r.t. $\alpha$. Recently, Balcan et al. (2025) provided the first *general framework* for the harder case where $\ell_x(\alpha)$ involves an inner optimization over model parameter $\theta$. However, as discussed, their results are restricted to one-dimensional hyperparameters and single-level objectives ($f \equiv g$). Our work directly extends this line of inquiry.

## 2. Preliminaries

### 2.1. Backgrounds on Learning Theory

We recall the notion of pseudo-dimension, a learning-theoretic complexity for a real-valued function class.

**Definition 2.1** (Pseudo-dimension, Pollard, 1984). Consider a real-valued function class $\mathcal{L} = \{\ell_\alpha : \mathcal{X} \to \mathbb{R} \mid \alpha \in \mathcal{A}\}$ parameterized by $\alpha \in \mathcal{A}$. Given a set of inputs $S = (x_1, \ldots, x_N) \subset \mathcal{X}$, we say that $S$ is shattered by $\mathcal{L}$ if there exists a set of real-valued threshold $t_1, \ldots, t_N \in \mathbb{R}$ such that $|\{(\text{sign}(\ell_\alpha(x_1) - t_1), \ldots, \text{sign}(\ell_\alpha(x_N) - t_N)) \mid \ell_\alpha \in \mathcal{L}\}| = 2^N$. The pseudo-dimension of $\mathcal{L}$, denoted $\text{Pdim}(\mathcal{L})$, is the maximum size $N$ of an input set that $\mathcal{L}$ can shatter.

A standard result in learning theory states that a real-valued function class with bounded pseudo-dimension is Probably Approximately Correct (PAC)-learnable with empirical risk minimization (ERM).

**Theorem 2.2** (Pollard, 1984). *Consider a real-valued function class* $\mathcal{L} = \{\ell_\alpha : \mathcal{X} \to [-H, H] \mid \alpha \in \mathcal{A}\}$ *parameterized by* $\alpha \in \mathcal{A}$. *Assume that* $\text{Pdim}(\mathcal{L})$ *is finite. Then given* $\epsilon > 0$ *and* $\delta \in (0, 1)$, *for any* $N \geq N(\epsilon, \delta)$, *where* $N(\epsilon, \delta) = \mathcal{O}\left(\frac{H^2}{\epsilon^2}(\text{Pdim}(\mathcal{L}) + \log(1/\delta))\right)$, *with probability at least* $1 - \delta$ *over the draw of* $S = (x_1, \ldots, x_N) \sim \mathcal{D}^N$, *where* $\mathcal{D}$ *is a distribution over* $\mathcal{X}$, *we have*

$$\mathbb{E}_{x \sim \mathcal{D}}[\ell_{\hat{\alpha}}(x)] \leq \inf_{\alpha \in \mathcal{A}} \mathbb{E}_{x \sim \mathcal{D}}[\ell_\alpha(x)] + \epsilon.$$

*Here* $\hat{\alpha} \in \arg\min_{\alpha \in \mathcal{A}} \sum_{x \in S} \ell_\alpha(x)$ *is the ERM minimizer w.r.t. the set of instances* $S$.

### 2.2. First-order Formula and Quantifier Elimination

In this section, we present the background necessary for quantifier elimination, a key component of our analysis. First, we introduce the notions of *first-order formula*.

**Definition 2.3** (First-order formula, quantified/free variables, and polynomial first-order logic (Renegar, 1992)). A **first-order formula** (FOL) $\Phi(\alpha)$ admits the form:

$$(Q_1 \theta^{[1]} \in \mathbb{R}^{d_1}) \ldots (Q_K \theta^{[K]} \in \mathbb{R}^{d_K}) P(\alpha, \theta^{[1]}, \ldots, \theta^{[K]}), \tag{2}$$

where

1. Each $Q_k$ is one of the quantifiers $\exists$ or $\forall$. The sequence $\{Q_k\}_{k=1}^K$ alternates between $\exists$ and $\forall$ and we denote $K$ as the number of quantifier alternations.

2. $\theta^{[1]}, \theta^{[2]}, \ldots, \theta^{[K]}$ are called the quantified variables, while $\alpha \in \mathbb{R}^p$ is called the free variable.

3. $P(\alpha, \theta^{[1]}, \theta^{[2]}, \ldots, \theta^{[K]})$ is a boolean combination of atomic predicates of the form:

$$P_j(\alpha, \theta^{[1]}, \theta^{[2]}, \ldots, \theta^{[K]}) \chi_j 0,$$

where $\chi_j \in \{>, \geq, <, \leq, =, \neq\}$ is relational operator.

A FOL $\Phi$ is a *polynomial* FOL if each $P_j$ is a polynomial of $\alpha$ and $\theta^{[1]}, \ldots, \theta^{[K]}$.

Essentially, a polynomial FOL is a logical statement built from three simple components: polynomial (in)equalities, Boolean operators (e.g., AND and OR), and quantifiers (e.g., 'there exists' $\exists$ and 'for all' $\forall$). The variables linked internally with the quantifiers (in our application, the parameters $\theta$ we optimize over) are called *quantified variables*, and the ones that are external inputs (in our application, the hyperparameters $\alpha$ we tune) are called *free variables*. We now introduce the notion of *quantifier-free formula*, which is an FOL that has no quantified variables.

**Definition 2.4** (Quantifier-free formula)**.** A polynomial FOL is a **quantifier-free formula** (QFF) if it has no quantified variables. In other words, it is a boolean combination of atomic predicates of the form $P_j(\alpha)\chi_j 0$, where $P_j$ is a polynomial of $\alpha \in \mathbb{R}^p$ and $\chi_j \in \{>, \geq, <, \leq, =, \neq\}$.

**Example 1.** We provide two examples of polynomial FOL for illustration purposes. Let $\alpha \in \mathbb{R}^p$ and $\theta \in \mathbb{R}^d$, then:

- The formula $\Phi(\alpha) = (\forall \theta \in \mathbb{R}^p)(\sum_{i=1}^{p} \alpha_i^2 - \sum_{j=1}^{d} \theta_j^2 \geq 0)$ is a polynomial FOL, where $\theta$ are quantified variables and $\alpha$ are free variables. In this case, there is only one quantifier alternation (e.g., $\forall \theta \in \mathbb{R}^n$), and therefore the number of quantifier alternations of $\Phi$ is $K = 1$. Furthermore, there is only one atomic predicate $P(\alpha, \theta) \geq 0$, where $P(\alpha, \theta) = \sum_{i=1}^{p} \alpha_i^2 - \sum_{j=1}^{d} \theta_j^2$ and since $P$ is a polynomial of $\alpha$ and $\theta$, $\Phi$ is a polynomial FOL.

- The formula $\Phi'(\alpha) = (\sum_{i=1}^{p} \alpha_i^2 - 1 \geq 0)$ is a quantifier-free polynomial FOL.

Finally, the following result asserts that we can transform a polynomial FOL to a polynomial QFF with a bounded number of atomic predicates and a maximum polynomial degree of those atomic predicates.

**Theorem 2.5** (Quantifier elimination algorithm, Basu et al. (2006, Algorithm 14.8))**.** *Given a polynomial FOL $\Phi$ as in (2) with a fixed number of quantifier alternation $K$, there exists an equivalent polynomial QFF $\Psi$ consisting of (polynomial) atomic predicates in the free variables $\alpha \in \mathbb{R}^p$. Let $M$ be the number of atomic predicates in $\Phi$, and $\Delta$ be their maximum degree. Then the formula $\Psi(\alpha)$ satisfies the following structural complexities:*

1. *Predicate complexity $I$: the number of distinct atomic polynomials appearing in $\Psi(\alpha)$ is at most*

$$I \leq M^{\prod_{k=1}^{K}(d_k+1)} \cdot \Delta^{\mathcal{O}(p) \cdot \prod_{k=1}^{K} d_k}.$$

2. *Degree complexity $\Delta_{QE}$: the degree of these atomic polynomials is at most*

$$\Delta_{QE} \leq \Delta^{\mathcal{O}(\prod_{k=1}^{K} d_k)}.$$

Later, we will see that the computation of a polynomial QFF can be described by a GJ algorithm, which is the key idea of our analysis.

## 3. Problem Settings

We follow the setting from Balcan et al. (2025) by considering a data-driven framework involving three fundamental spaces: (1) a problem instance space $\mathcal{X} \subseteq \mathbb{R}^q$, (2) a hyperparameter space $\mathcal{A} \subseteq \mathbb{R}^p$, and (3) a model parameter space $\Theta \subseteq \mathbb{R}^d$. For simplicity, we assume $\mathcal{A} = [\alpha_{\min}, \alpha_{\max}]^p$ and $\Theta = [\theta_{\min}, \theta_{\max}]^d$. For any problem instance $x \in \mathcal{X}$, we evaluate the performance of a model parameterized by $\theta \in \Theta$ and hyperparameters $\alpha \in \mathcal{A}$ using two distinct objective functions:

- *Training objective $f : \mathcal{X} \times \mathcal{A} \times \Theta \to [-H, H]$:* the surrogate loss minimized by the algorithm, and

- *Validation objective $g : \mathcal{X} \times \mathcal{A} \times \Theta \to [-H, H]$:* the target metric used to evaluate the quality of the solution.

Here, $H$ is a threshold for boundedness. The induced loss $\ell_\alpha(x)$ measures the performance of the model on the problem instance $x$ given hyperparameters $\alpha$, defined via the bi-level structure in (1). Notice that we adopt the *optimistic bi-level optimization* formulation (Dempe, 2002), which is standard in the literature to ensure the validation is well-defined, then the lower-level set $\mathcal{S}(x, \alpha)$ is not a singleton.

**Objectives.** We assume there is an application-specific, unknown problem distribution $\mathcal{D}$ over $\mathcal{X}$. The goal of data-driven hyperparameter tuning is to find a value $\alpha^*$ that minimizes the expected loss

$$\alpha^* \in \arg\min_{\alpha \in \mathcal{A}} \mathbb{E}_{x \sim \mathcal{D}}[\ell_\alpha(x)].$$

Since $\mathcal{D}$ is unknown, we observe $N$ training instances $S = \{x_1, \ldots, x_N\} \sim \mathcal{D}^N$ and minimize the empirical risk

$$\hat{\alpha}(S) \in \arg\min_{\alpha \in \mathcal{A}} \frac{1}{N} \sum_{i=1}^{N} \ell_\alpha(x_i).$$

Our primary theoretical goal is to establish sample complexity guarantees for this learning problem. Exploiting results Theorem 2.2, it suffices to study the induced function class $\mathcal{L} = \{\ell_\alpha : \mathcal{X} \to [-H, H] \mid \alpha \in \mathcal{A}\}$, and bound its pseudo-dimension $\mathrm{Pdim}(\mathcal{L})$.

**Structural assumption.** We focus on the case where $f$ and $g$ admits *piecewise polynomial structure*, formally defined as follows.

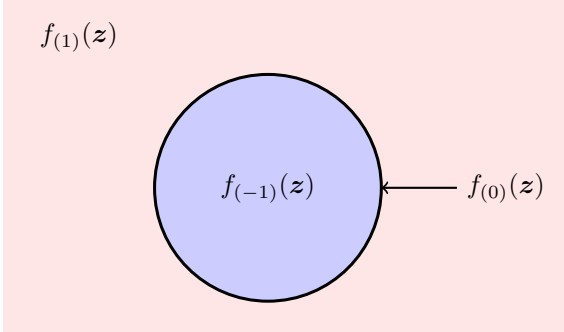

*Figure 1.* A simple illustration of the piecewise polynomial structure. Here, the set of boundary polynomials $\mathbb{H} = \{h_1\}$, where $h_1(z) = z_1^2 + z_2^2 - 4$. In the region $\{z \in \mathbb{R}^2 \mid h_1(z) > 0\}$ (i.e, the region outside the circle, the sign pattern $\boldsymbol{\sigma}(z) = (1) \in \{-1, 0, 1\}^1$), the function $f(z)$ admits the polynomial form $f_{(-1)}(z) = z_1 - z_2$. Similarly, in the region $\{z \in \mathbb{R}^2 \mid h_1(z) < 0\}$ inside the circle (i.e., the sign pattern $\boldsymbol{\sigma}(z) = (-1)$), we have $f(z) = f_{(-1)}(z) = z_1 + z_2$. Finally, in the region where $\{z \in \mathbb{R}^2 \mid h_1(z) = 0\}$ (i.e., the boundary of the circle, $\boldsymbol{\sigma}(z) = (0)$), $f(z) = f_{(0)}(z) = z_1^3$. Note that all pieces are polynomials in $z$, and the complexity of the function $f(z)$ is $(1, 3, \Delta)$, where $\Delta = \max\{\deg(h_1), \deg(f_{(0)}), \deg(f_{(-1)}), \deg(f_{(1)})\} = 3$.

**Definition 3.1** (Piecewise polynomial function (Balcan et al., 2025))**.** A real-valued function $f$ admits a *piecewise polynomial structure* with complexity $(M_f, T_f, \Delta_f)$ if there exists a set of boundary polynomials $\mathbb{H} = \{h_1, \ldots, h_{M_f}\}$ and a set of value polynomials $\mathbb{F} = \{f_{\boldsymbol{\sigma}}\}_{\boldsymbol{\sigma} \in \Sigma_f} \subset \mathbb{R}[z]$, where $\Sigma_f \subseteq \{-1, 0, 1\}^{M_f}$ and $|\Sigma_f| \leq T_f$, both with polynomial degrees at most $\Delta$, such that for any $z$, we have $f(z) = f_{\boldsymbol{\sigma}(z)}(z)$, where $\boldsymbol{\sigma}(z) \in \Sigma_f$ is the sign pattern of $z$ with respect to the set of functions $\mathbb{H}$, i.e., $\boldsymbol{\sigma}(z)_j = \text{sign}(h_j(z))$. The functions in $\mathbb{F}$ are called the *piece functions* and the functions in $\mathbb{H}$ are called the *boundary functions*.

**Assumption 3.2** (Structural assumption)**.** Given any problem instance $x$, the dual parameter-dependent training objective $f_x(\alpha, \theta) \triangleq f(x, \alpha, \theta)$ and dual parameter-dependent validation objective $g_x(\alpha, \theta) \triangleq g(x, \alpha, \theta)$ admits piecewise polynomial structure (as functions of $\alpha$ and $\theta$) with complexity $(M_f, T_f, \Delta_f)$ and $(M_g, T_g, \Delta_g)$ (independent of the problem instance $x$), respectively. Moreover, the set $\mathcal{S}(x, \alpha)$ of minimizes of $f(x, \alpha, \cdot)$ is non-empty, for all $(x, \alpha)$.

*Remark* 3.3 (Ubiquity of structure)**.** As noted by Balcan et al. (2025), this structural assumption is robust and ubiquitous. It has been established for a wide range of applications in both classical learning theory (Bartlett et al., 1998; Montúfar et al., 2014; Bartlett et al., 2019) and data-driven algorithm design problems (Balcan et al., 2021a; 2023; Nguyen & Nguyen, 2026; Cheng & Basu, 2025).

While Balcan et al. (2025) initiated the formal study of this setting, their results are restricted to one-dimensional hyperparameter case ($\mathcal{A} \subset \mathbb{R}$) and the single-level setting

($f \equiv g$). We now present our general results, which fully eliminate these two restrictions.

# 4. A General Learning-theoretic Complexity Framework via First-Order Logic

In this section, we present a general result that bounds the pseudo-dimension of a function class $\mathcal{F} = \{f_\alpha : \mathcal{X} \to [-H, H] \mid \alpha \in \mathcal{A} \subseteq \mathbb{R}^p\}$. The next theorem serves as the engine of our analysis, converting logical definability into statistical learnability. Its proof is presented in Appendix B.

**Theorem 4.1** (Pseudo-dimension bound)**.** *Consider a class of functions $\mathcal{F} = \{f_\alpha : \mathcal{X} \subseteq \mathbb{R}^q \to \mathbb{R} \mid \alpha \in \mathcal{A}\}$, where $\mathcal{A} = [\alpha_{\min}, \alpha_{\max}]^p \subset \mathbb{R}^p$. Suppose that for any fixed instance $x \in \mathcal{X}$ and threshold $t \in \mathbb{R}$, the boolean value $\mathbb{I}(f_\alpha(x) \geq t)$ is equivalent to a polynomial FOL $\Phi_{x,t}(\alpha)$ (cf. Definition 2.3) with a fixed quantifier alternation $K \in \mathbb{N}$. Assume that for any problem instance $x$ and threshold $t$:*

- *$\Phi_{x,t}$ involves $K$ blocks of quantifiers with variable dimensions $(d_1, \ldots, d_K)$,*

- *The number of polynomials appearing in $\Phi_{x,t}$ and their maximum degree are uniformly bounded by $M$ and $\Delta$,*

*Then the pseudo-dimension $\text{Pdim}(\mathcal{F})$ is upper-bounded by*

$$O\left(p \prod_{k=1}^{K}(d_k + 1) \log M + p^2 \prod_{k=1}^{K} d_k \log \Delta\right).$$

**Comparison to existing works.** To the best of our knowledge, Goldberg & Jerrum (1993) is the only work in the literature studying the pseudo-dimension of the polynomial FOLs. Goldberg & Jerrum (1993) gave a bound [1]

$$O\left(p(p + q) \prod_{k=1}^{K} d_k \left(\log M + \log \Delta\right)\right), \qquad (3)$$

and our Theorem 4.1 is finer than this bound. In particular, we remove the dependency on the data dimension $q$ and a factor $p$ in front of $\log M$. This improvement is significant, for example, when the function class contains piecewise polynomial functions with exponential pieces/boundaries, or when the data dimension is larger than the number of tuning parameters, $q \gg p$.

Theorem 4.1 applies to many practical learning situations. The only condition we need to verify is that the objective functions $f$ and $g$ can be described as polynomials and/or polynomial FOLs. The class of functions that admit description with polynomials is precisely the set of *semi-algebraic*

---

[1]Comparing to original statement, the bound reported in (3) is missing a factor $2^{O(K)}$. This is because when $K$ is fixed, this constant is absorbed in the big O notation.

functions. We provide interested readers with more details about semi-algebraic functions in Appendix C. In fact, many functions used in the learning context are semi-algebraic because this class does not only contain piecewise polynomial functions (cf. Definition 3.1), but also many others such as $\|x\|_p, p \in \mathbb{N}, p \geq 2$ or well-known sparse regularisations such as $\ell_2/\ell_1$ ratio, group LASSO, which are not instances of Definition 3.1 (also see Appendix C for more details). Therefore, the scope of Theorem 4.1 can go up to the set of semi-algebraic functions.

In Section 5 and Section 6, we demonstrate applications of Theorem 4.1 for $f$ and $g$ being piecewise polynomial functions and address an open question posed in (Balcan et al., 2025). In addition, we also demonstrate an example of a learning problem whose functions are semi-algebraic but not covered by Section 5 and Section 6: hyper-parameter tuning for the weighted group LASSO in Section 8.1.

## 5. Data-driven Tuning via Training Objective

### 5.1. Upper bound

In this section, we first analyze the setting where the hyperparameter $\alpha$ is tuned to minimize the training objective directly (i.e., the case where $f \equiv g$). In this scenario, for a fixed problem instance $x$ and hyperparameter $\alpha$, the loss $\ell_\alpha(x)$ is defined implicitly as the optimal value of the training problem:

$$\ell_\alpha(x) = \min_{\theta \in \Theta} f(x, \alpha, \theta).$$

Recall that from Assumption 3.2, for any given problem instance $x$, $f_x(\alpha, \theta) \triangleq f(x, \alpha, \theta)$ admits piecewise polynomial structure (Definition 3.1) with complexity $(M_f, T_f, \Delta_f)$. Under such an assumption, the following result establishes a pseudo-dimension upper-bound for the function class $\mathcal{L}$ induced by $\ell_\alpha : \mathcal{X} \to [-H, H]$ when varying $\alpha \in \mathcal{A} = [\alpha_{\min}, \alpha_{\max}]^p \subset \mathbb{R}^p$.

**Theorem 5.1** (Pseudo-dimension – Training loss). *Let $\mathcal{A} = [\alpha_{\min}, \alpha_{\max}]^p$ and $\Theta = [\theta_{\min}, \theta_{\max}]^d$. Suppose that for any instance $x$, the training objective $f_x(\alpha, \theta) \triangleq f(x, \alpha, \theta)$ for $(\alpha, \theta) \in \mathcal{A} \times \Theta$ admits a piecewise polynomial structure (cf. Definition 3.1) with complexity $(M_f, T_f, \Delta_f)$. Then, the pseudo-dimension of the function class $\mathcal{L} = \{\ell_\alpha : \mathcal{X} \to [-H, H] \mid \alpha \in \mathcal{A}\}$, where $\ell_\alpha(x) = \min_{\theta \in \Theta} f(x, \alpha, \theta)$, is bounded by*

$$\mathrm{Pdim}(\mathcal{L}) = \mathcal{O}(pd \log(M_f + T_f + d) + p^2 d \log \Delta_f).$$

*Proof.* To apply Theorem 4.1, given a problem instance $x$ and a real-valued threshold $t$, our goal is to construct a polynomial FOL formula $\Phi_{x,t}(\alpha)$ equivalent to $\mathbb{I}(\ell_\alpha(x) \geq t)$. Since $\ell_\alpha(x) = \min_{\theta \in \Theta} f_x(\alpha, \theta)$ is a minimization over $\Theta$, the condition $\ell_\alpha(x) \geq t$ is equivalent to stating that for

all parameter $\theta \in \Theta$, the function value $f_x(\alpha, \theta)$ is greater or equal than $t$, and $\Phi_{x,t}(\alpha)$ is defined as

$$\Phi_{x,t}(\alpha) \triangleq (\forall \theta \in \mathbb{R}^d)[(\theta \in \Theta) \Rightarrow f_x(\alpha, \theta) \geq t]$$
$$= (\forall \theta \in \mathbb{R}^d)[\neg(\theta \in \Theta) \vee (f_x(\alpha, \theta) \geq t)].$$

Here, we use the logical identity $(A \Rightarrow B) = (\neg A \vee B)$ (i.e., not $A$ or $B$). It is now sufficient to apply Theorem 4.1. Details are in Appendix D.1. □

Combining Theorem 5.1 with Theorem 2.2, we could establish the sample complexity guarantee for multi-dimensional hyperparameter tuning using the training loss function. Our result generalizes the guarantees of Balcan et al. (2025) from one-dimensional hyperparameters to the multi-dimensional case, where $p > 1$. Moreover, our logic-based analysis avoids the need for strong regularity assumptions on the boundary polynomials $\{h_{x,i}\}_{i=1,\dots,M_f}$ (see e.g., Balcan et al. (2025, Assumption 1)), which were required by the geometric approach to prevent topological pathologies in the solution path.

### 5.2. Lower bound

In this section, we will instantiate the *first* general lower bound for the pseudo-dimension in such a setting. The proof idea originates from the *bit-extraction technique* (Bartlett et al., 2019), but requires a *novel stabilization argument* to handle the implicit optimization problem in the definition of the function class.

**Theorem 5.2** (Pseudo-dimension lower bound - Training loss). *Let $\mathcal{A} = \mathbb{R}^p$, $\Theta = \mathbb{R}^d$. Then for every sufficiently large $\Delta_f > 0$, there exists a function class $\mathcal{L} = \{\ell_\alpha : \mathcal{X} \to \mathbb{R} \mid \alpha \in \mathcal{A}\}$, where $\ell_\alpha(x) = \min_{\theta \in \Theta} f(x, \alpha, \theta)$, and $f(x, \alpha, \theta)$ is a degree at most $\Delta_f$ for any problem instance $x \in \mathcal{X}$, such that $\mathrm{Pdim}(\mathcal{L}) = \Omega(pd \log \Delta_f)$.*

*Proof.* It suffices to show that there exists $N = \Omega(pd \log \Delta_f)$ problem instances $x_1, \dots, x_N$ and $N$ real-valued thresholds $\tau_1, \dots, \tau_N \in \mathbb{R}$ such that for any bit vector $y \in \{0, 1\}^N$ there exists a hyperparameter $\alpha_y \in \mathcal{A}$ such that $\mathbb{I}(\ell_{\alpha_y}(x_t) - \tau_t \geq 0) = y_t$ for any $t = 1, \dots, N$. We will construct the problem instances $x_t$ and the parameterized function class $\mathcal{L} = \{\ell_\alpha : \mathcal{X} \to \mathbb{R} \mid \alpha \in \mathcal{A}\}$ as follows:

- The construction of $x_t$: Let $K = \lfloor \Delta_f/2 \rfloor$, $B = \lfloor \log_2 K \rfloor$. Let $N = p \cdot d \cdot B$, then it is obvious that $N = \Omega(pd \log \Delta_f)$. For the triplet $(j, i, b)$, where $j \in \{1, \dots, p\}$, $i \in \{1, \dots, d\}$, and $b \in \{1, \dots, B\}$, we define the problem instance $x^{(j,i,b)}$ as the tuple of one hot vectors $(u, v, w) \in \{0, 1\}^{p \times d \times B}$. Here $u_j = v_i = w_b = 1$, and all other entries are 0.

- The construction of $\ell_\alpha(x)$ : Since $\ell_\alpha(x) = \min_{\theta \in \Theta} f(x, \alpha, \theta)$, it suffices to construct $f(x, \alpha, \theta)$, which is

$$f(x, \alpha, \theta) = C \sum_{m=1}^{d} \prod_{k=0}^{K-1} (\theta_m - k)^2$$
$$+ \left( \sum_{n=1}^{p} u_n \alpha_n - \sum_{m=1}^{d} \theta_m K^{m-1} \right)^2$$
$$+ 0.5 \sum_{m=1}^{d} \sum_{c=1}^{B} v_m w_c E_c(\theta_m).$$

Here $E_c(t) = \sum_{j=0}^{K-1} \beta_{j,c} \left( \prod_{\substack{m=0 \\ m \neq j}}^{K-1} \frac{t-m}{j-m} \right)$ is the *bit-extracting polynomial*, and $C$ is a sufficiently large constant that is independent on $y, j, i, b$.

Based on that construction, we can claim that:

1. $\ell^{\mathcal{K}}(\alpha_y)(x^{(j,i,b)}) = \frac{y^{(j,i,b)}}{2}$ for any $y \in \{0,1\}^{p \times d \times B}$, for some $\alpha_y \in \mathcal{A}$ depending on $y$, and $\ell^{\mathcal{K}}(\alpha)(x) = \min_{\theta \in \mathcal{K}^d} f(x, \alpha, \theta)$, $\mathcal{K} = \{0, 1, \ldots, K-1\}$.

2. $\ell_{\alpha_y}(x^{(j,i,b)}) \in (\ell^{\mathcal{K}}_{\alpha_y}(x^{(j,i,b)}) - 0.1, \ell^{\mathcal{K}}_{\alpha_y}(x^{(j,i,b)}))$.

Finally, simply choose $\tau_t = 0.25$, and we have the final conclusion. The details are provided in Appendix D.2. □

# 6. Data-driven Tuning via Validation Objective

We now address the general data-driven setting in which the hyperparameter $\alpha$ is tuned to minimize a validation objective $g$ evaluated on the optimal training parameters. As noted earlier, this formulation is general and can capture standard practices in hyperparameter tuning (e.g., tuning a regularization coefficient to minimize validation loss in LASSO). In this scenario, for a fixed problem instance $x$ and hyperparameter $\alpha$, the loss $\ell_\alpha(x)$ is defined as $\ell_\alpha(x) = \inf_{\theta \in \mathcal{S}(x,\alpha)} g_x(\alpha, \theta)$, where $\mathcal{S}(x, \alpha) = \arg\min_{\theta \in \Theta} f_x(\alpha, \theta)$.

Recall from Assumption 3.2 that for any fixed problem instance $x$, $f_x(\alpha, \theta)$ and $g_x(\alpha, \theta)$ admits piecewise polynomial structure with complexity $(M_f, T_f, \Delta_f)$ and $(M_g, T_g, \Delta_g)$, respectively. Under this assumption, we obtain the following result, establishing the learning guarantee for data-driven hyperparameter tuning with a validation objective.

**Theorem 6.1** (Pseudo-dimension – Validation loss)**.** *Let $\mathcal{A} = [\alpha_{\min}, \alpha_{\max}]^p$ and $\Theta = [\theta_{\min}, \theta_{\max}]^d$. Suppose that for any instance $x$, the training objective $f_x(\alpha, \theta)$ and the validation objective $g_x(\alpha, \theta)$, for $(\alpha, \theta) \in \mathcal{A} \times \Theta$, admit piecewise polynomial structures (cf. Definition 3.1) with complexity $(M_f, T_f, \Delta_f)$ and $(M_g, T_g, \Delta_g)$, respectively. Then, the*

*pseudo-dimension of the function class $\mathcal{L} = \{\ell_\alpha : \mathcal{X} \to [-H, H] \mid \alpha \in \mathcal{A}\}$, where $\ell_\alpha(x) = \inf_{\theta \in \mathcal{S}(x,\alpha)} g_x(\alpha, \theta)$ and $\mathcal{S}(x, \alpha) = \arg\min_{\theta \in \Theta} f_x(\alpha, \theta)$, is bounded by*

$$\mathrm{Pdim}(\mathcal{L}) = \mathcal{O}(pd^2 \log M_{\mathrm{tot}} + p^2 d^2 \log \Delta_{\mathrm{tot}}),$$

*where $M_{\mathrm{tot}} = M_f + T_f + M_g + T_g + d$, and $\Delta_{\mathrm{tot}} = \max(\Delta_f, \Delta_g)$.*

*Proof.* Similar to the proof of Theorem 5.1, the idea is to use Theorem 4.1 by showing that: given a problem instance $x$ and a real-valued threshold $t$, there is a polynomial FOL $\Phi_{x,t}(\alpha)$ equivalent to $\mathbb{I}(\ell_x(\alpha) \geq t)$ with bounded complexities. Such a polynomial FOL can be defined as follows:

$$\Phi_{x,t}(\alpha) \triangleq (\forall \theta \in \mathbb{R}^d) [(\theta \in \mathcal{S}(x, \alpha)) \Rightarrow (g(x, \alpha, \theta) \geq t)]$$
$$= (\forall \theta \in \mathbb{R}^d) [\neg(\theta \in \mathcal{S}(x, \alpha)) \vee (g(x, \alpha, \theta) \geq t)].$$

Here, we again use the identity $(A \Rightarrow B) = (\neg A \vee B)$. We first expand the optimality constraints $\theta \in \mathcal{S}(x, \alpha)$. Note that a parameter $\theta$ is not optimal (i.e., $\neg(\theta \in \mathcal{S}(x, \alpha))$ if it is not in the region $\Theta$ or if there exists a better candidate $\theta'$ such that $f_x(\alpha, \theta') < f_x(\alpha, \theta)$. Therefore, $\neg(\theta \in \mathcal{S}(x, \alpha))$ can be rewritten as

$$(\theta \notin \Theta) \vee [(\exists \theta' \in \mathbb{R}^d)[(\theta' \in \Theta) \wedge (f_x(\alpha, \theta') < f_x(\alpha, \theta))]].$$

Let $L_1 = (\theta' \in \Theta) \wedge (f_x(\alpha, \theta') < f_x(\alpha, \theta))$, which is the logical sentence for *optimality check*, we can then write $\Phi_{x,t}(\alpha)$ as

$$(\forall \theta \in \mathbb{R}^d)(\exists \theta' \in \mathbb{R}^d) \left[ \underbrace{\theta \notin \Theta}_{\text{Domain check}} \vee \underbrace{(g_x(\alpha, \theta) \geq t)}_{\text{Validation check}} \vee L_1 \right].$$

□

Combining Theorem 6.1 with Theorem 2.2, we could establish the sample complexity guarantee for multi-dimensional hyperparameter tuning using the validation loss function.

## 6.1. Handling approximate lower-level optimization

Definition of $\mathcal{S}(x, \alpha) = \arg\min_{\theta \in \Theta} f_x(\alpha, \theta)$ involves in the *exact minimization*, which might be too strict in practice. In this section, we demonstrate that our proposed framework can flexibly handle the case of *inner approximate minimization*, where $\theta$ is in the $\epsilon$-approximate optimal set $S_\epsilon(x, \alpha) = \{\theta \in \Theta \mid f(x, \alpha, \theta) \leq \inf_{\theta' \in \Theta} f(x, \alpha, \theta') + \epsilon\}$. We define the loss as $\ell^\epsilon_\alpha(x) = \inf_{\theta \in \mathcal{S}_\epsilon(x,\alpha)} g(x, \alpha, \theta)$. Under such notions, we can establish the pseudo-dimension upper-bound as follows.

**Proposition 6.2.** *Under the same assumptions as Theorem 5.1, for every fixed $\epsilon > 0$, define the function class $\mathcal{L}_\epsilon = \{\ell^\epsilon_\alpha : \mathcal{X} \to [-H, H] \mid \alpha \in \mathcal{A}\}$. Then $\mathrm{Pdim}(\mathcal{L}_\epsilon) = \mathcal{O}(pd^2 \log M_{\mathrm{tot}} + p^2 d^2 \log \Delta_{\mathrm{tot}})$.*

*Proof.* The key idea is that

$$\ell_\alpha^\epsilon(x) \geq t \Leftrightarrow (\forall \theta \in S_\epsilon(x,\alpha))[g(x,\alpha,\theta) \geq t],$$

or equivalently

$$(\forall \theta \in \mathbb{R}^d)[\theta \in S_\epsilon(x,\alpha) \Rightarrow g(x,\alpha,\theta) \geq t].$$

After expanding the approximate optimality, the threshold predicate can be written as

$$(\forall \theta \in \mathbb{R}^d)(\exists \theta' \in \mathbb{R}^d)$$
$$[(\theta \notin \Theta) \vee (g(x,\alpha,\theta) \geq t)$$
$$\vee (\theta' \in \Theta \wedge f(x,\alpha,\theta) > f(x,\alpha,\theta') + \epsilon)].$$

Finally, applying Theorem 4.1, we have the conclusion. $\square$

# 7. Refined Sample Complexity Bounds via Explicit Solution Paths

The general framework established in Sections 5 and 6 provides general guarantees for implicitly defined loss functions. However, in many practical settings, such as LASSO and ridge regression (Tibshirani, 1996; Hoerl & Kennard, 1970), the inner optimization problem is not a black box. Instead, it admits an explicit analytical structure, typically as a unique *solution path* $\theta^*(x,\alpha)$ that admits a piecewise structure to $\alpha$. In this section, we demonstrate that our model-theoretic perspective can exploit this additional structure to derive significantly tighter bounds. By directly analyzing the complexity of the solution path, we can bypass the quantifier elimination step, often removing the dependence on the parameter dimension $d$, and, in some scenarios, recover the sample complexity rate that matches known lower bounds. We formalize this structural assumption as follows.

**Assumption 7.1.** Given a problem instance $x$, we assume that $\alpha \mapsto \theta^*(x,\alpha) = \arg\min_{\theta \in \Theta} f(x,\alpha,\theta)$ is a piecewise rational function with complexity $(M_{\text{path}}, T_{\text{path}}, \Delta_{\text{path}})$. Specifically:

- There exists a set of at most $M_{\text{path}}$ boundary rational functions $\mathbf{H}_x = \{h_{x,1}, \ldots, h_{x,M_{\text{path}}}\}$ of $\alpha$.

- For any $\alpha \in \mathcal{A}$, the set of minimizer $\arg\min_{\theta \in \Theta} f(x,\alpha,\theta) = \{\theta^*(x,\alpha)\}$ has an unique element. Moreover, the optimal parameter is given by $\theta^*(x,\alpha) = \theta_{\sigma(\alpha)}(\alpha)$, where $\sigma(\alpha) \in \Sigma_{x,\text{path}} \subset \{-1,0,1\}^{M_{\text{path}}}$ is the sign pattern of $\alpha$ with respect to the set of boundary $\mathbf{H}_{x,i}$, i.e., $\sigma(\alpha)_i = \text{sign}(h_{x,i}(\alpha))$, and $\theta^*_{\sigma(\alpha)}(x,\alpha)$ is a rational function of $\alpha$.

- The set of sign pattern $\Sigma_{x,\text{path}}$ has at most $T_{\text{path}}$ elements, i.e., $|\Sigma_{x,\text{path}}| \leq T_{\text{path}}$. Besides, all rational functions $h_{x,i}$ and $\theta_{\sigma(\alpha)}(\alpha)$ are of degree at most $\Delta_{\text{path}}$.

Assumption 7.1 implies the following: First, unlike the general setting where $\mathcal{S}(x,\alpha)$ is implicitly defined via logical predicates, Assumption 7.1 guarantees an explicit function form $\theta^*(x,\alpha)$, allowing us to bypass the heavy machinery of quantifier elimination. Second, under Assumption 7.1, the distinction between the case $f \equiv g$ and $f \not\equiv g$ effectively vanishes. Since $\theta^*(x,\alpha)$ is explicit, the loss becomes $\ell_\alpha(x) = g(x,\alpha,\theta^*(x,\alpha))$. Both settings reduce to analyzing a single composite piecewise rational function, eliminating the extra quantifier block typically required for validation constraints.

Under a refined structure in Assumption 7.1, we can establish improved generalization guarantees for learning the function classes $\mathcal{L}$.

**Theorem 7.2** (Improved guarantee). *Given a problem instance $x$, suppose Assumption 7.1 holds for the optimal parameters $\theta^*(\alpha, w)$, and the tuning objective $g_x(\alpha, \theta)$ (which is $f_x(\alpha, \theta)$ if $f \equiv g$ or $g_x(\alpha, \theta)$ if $f \not\equiv g$) admits a piecewise rational structure with complexity $(M_k, T_k, \Delta_k)$. Then $\text{Pdim}(\mathcal{L}) = \mathcal{O}(p\log(M_{\text{total}}\Delta_{\text{total}}))$, where $M_{\text{total}} = M_{\text{path}} + T_{\text{path}} \cdot (M_k + T_k)$ and $\Delta_{\text{total}} = \Delta_k \cdot \Delta_{\text{path}}$.*

*Proof sketch.* In this scenario, we can apply the GJ framework directly without invoking quantifier elimination. Besides, since $\theta^*(x,\alpha)$ becomes a rational function of $\alpha$, we expect that the final bound should only depend on the dimensionality of $\alpha$ instead of $\theta$. Recall that, to give an upper-bound for the pseudo-dimension of $\mathcal{L}$ using GJ framework, for any problem instance $x \in \mathcal{X}$ and and any real-valued threshold $t \in \mathbb{R}$, we want that the computation of $\mathbb{I}(\ell_x(\alpha) \geq t)$, where $\ell_\alpha(x) = k_x(\alpha, \theta^*(x,\alpha))$, can be described by a GJ algorithm (cf. Definition A.1) with bounded complexities. Finally, we describe such computation, using the piecewise rational structure of $\theta^*(x,\alpha)$ and $k_x(\alpha,\theta)$, which gives us the final guarantees. See Appendix F for the detailed proof. $\square$

We note that under Assumption 7.1, the bound given by Theorem 7.1 is *tight in some scenarios*. Concretely, there exists a problem (e.g., data-driven tuning of the elastic-net regularization hyperparameters (Balcan et al., 2023)) such that the upper-bound for the pseudo-dimension from Theorem 7.2 matches the lower-bound presented in Balcan et al. (2023). See Appendix F.2 for a detailed discussion.

# 8. Applications in Learning Problems

In this section, we demonstrate applications of Theorem 4.1 in two hyperparameter tuning problems: Weight Group Lasso, which violates the piecewise polynomial assumption of Assumption 3.2, and Weighted Fused Lasso, which has a better pseudo-dimension estimation.

## 8.1. Data-driven Weighted Group Lasso - Beyond Piecewise Polynomial Assumptions

As discussed after Theorem 4.1, our machinery goes beyond piecewise polynomial structures. In this section, we demonstrate this with the weighted group Lasso sparse regularization (Yuan & Lin, 2006). In particular, we consider problem (1) with:

$$\overbrace{}^{\text{weighted group LASSO}}$$

$$f(x, \alpha, \theta) = \|A\theta - b\|^2 + \sum_{i=1}^{p} \alpha_i \|\theta_i\|_2 , \quad (4)$$

$$g(x, \alpha, \theta) = \|A'\theta - b'\|^2,$$

where $x = (A, A', b, b')$, $\theta = (\theta_1, \ldots, \theta_p) \in \mathbb{R}^d$ and $\alpha \in \mathbb{R}^p$. No constraint is imposed on $\theta$, i.e., $\Theta = \mathbb{R}^d$.

It is noteworthy that the use of the weighted group LASSO regularizer makes $f$ no longer piecewise polynomial (see Appendix C), and previous results are unable to handle this case. Nevertheless, Theorem 4.1 still leads to an upper bound for the pseudo-dimension.

**Theorem 8.1** (Pseudo-dimension for weighted group lasso). *Consider the class of functions $\mathcal{L} = \{\ell_\alpha \mathcal{X} \to [-H, H] \mid \alpha \in \mathcal{A}\}$ where $\ell_\alpha$ is defined as in (1) and $f, g$ are defined as in (4). We have:*

$$\text{Pdim}(\mathcal{L}) = \mathcal{O}(p^3 d + p^2 d^2).$$

*Proof sketch.* To use Theorem 4.1, we can add additional variables to represent $f$ with polynomials. Indeed, with extra scalar variables $\nu_1, \ldots, \nu_p$, we have:

$$f(x, \alpha, \theta) = \|A\theta - b\|^2 + (\alpha_1 \nu_1 + \ldots \alpha_p \nu_p),$$

with polynomial constraints:

$$\nu_i^2 = \sum_j [\theta_i]_j^2 \quad \text{and} \quad \nu_i \geq 0 \quad \forall i = 1, \ldots, p.$$

The detailed proof is presented in Appendix G. □

## 8.2. Data-driven Weighted Fused Lasso

We consider the problem of signal denoising with structural change-point detection, commonly modeled using the *Weighted Fused LASSO* (Tibshirani et al., 2005). Given a noisy signal $b$, the goal is to recover an approximation $\theta \in \mathbb{R}^d$. While standard Fused LASSO uses simple scalar regularization parameters, practical applications often benefit from allowing spatially varying regularization weights $\alpha \in \mathbb{R}^{d-1}$ to capture the heterogeneous noise levels or varying structural density across the signal.

In particular, we consider Problem (1) with:

$$f(x, \alpha, \theta) = \frac{1}{2}\|b - A\theta\|_2^2 + \sum_{i=1}^{d-1} \alpha_i |\theta_{i+1} - \theta_i| ,$$

$$(5)$$

$$g(x, \alpha, \theta) = \frac{1}{2}\|A'\theta - b'\|^2,$$

where $x = (A, A', b, b')$, $\alpha \in \mathbb{R}^p$ with $p = d - 1$, $\theta \in \mathbb{R}^d$. No constraint is imposed, i.e., $\Theta = \mathbb{R}^d$. In particular, we assume that the data distribution of $x$ satisfies that the matrix $A$ always has full column rank. We have the next result.

**Theorem 8.2** (Pseudo-dimension for weighted fused lasso). *Consider the class of functions $\mathcal{L} = \{\ell_\alpha \mathcal{X} \to [-H, H] \mid \alpha \in \mathcal{A}\}$ where $\ell_\alpha$ is defined as in (1) and $f, g$ are defined as in (5). We have:*

$$\text{Pdim}(\mathcal{L}) = \mathcal{O}(d^2).$$

The proof exploits Theorem 7.2; the details are in Appendix G.

# 9. Conclusion and Future Works

Our paper established a general learning-theoretic framework for data-driven hyperparameter tuning with implicitly defined loss functions. By bridging statistical learning theory and model theory, we derived the first general sample-complexity guarantees for multi-dimensional hyperparameter tuning ($\alpha \in \mathbb{R}^p$), thereby resolving a key open question in the existing literature. Additionally, we establish the first lower bound construction for this general setting. Furthermore, we demonstrated that by leveraging additional algebraic structure and explicit solution paths, we can bypass the general worst-case analysis to achieve significantly tighter bounds. Finally, we illustrated the versatility of our framework through new applications in sparse learning and robust optimization.

Our work opens two primary questions for future research. First, the fundamental lower bounds for the general bi-level setting remain unknown. Although we established matching lower bounds for specific cases with explicit solution paths (Section 7), determining the minimax lower bound for the general implicitly defined setting is an important open problem. Second, our current framework relies on semi-algebraic geometry. A natural extension is to generalize this logic-based approach to *o-minimal structures* (Van den Dries, 1998), such as Pfaffian functions. This would extend our guarantees to a much broader class of machine learning objectives involving transcendental functions like $\exp$, $\log$, and $\tanh$, moving beyond the polynomial limitations of the current work.

## Acknowledgment

Research reported in this paper was partially supported through the French ANR through the MIAI Cluster (reference ANR-23-IACL-0006). Viet Anh Nguyen gratefully acknowledges the support from the CUHK's Improvement on Competitiveness in Hiring New Faculties Funding Scheme, UGC ECS Grant 24210924, and UGC GRF Grant 14208625.

## Impact Statement

This paper presents work whose goal is to advance the field of AI and Machine Learning. There are many potential societal consequences of our work, none which we feel must be specifically highlighted here.

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

# A. The Goldberg-Jerrum (GJ) Framework

The Goldberg-Jerrum (GJ) framework was originally proposed by Goldberg & Jerrum (1993), and later refined by Bartlett et al. (2022). It establishes the pseudo-dimension upper-bound for parameterized function class $\mathcal{L}$, of which the computation of each function $\ell_\alpha$ can be described by an *GJ algorithm* using basic operators $(+, -, \times, \div)$, conditional statements, and intermediate values which are typically rational functions (e.g., fractions of two polynomials) of $\alpha$. The formal definition of the GJ framework is as follows.

**Definition A.1** (GJ algorithm, (Bartlett et al., 2022)). A GJ algorithm $\Gamma$ operates on real-valued inputs, and can perform two types of operations:

- Arithmetic operators of the form $v'' = v \odot v'$, where $\odot \in \{+, -, \times, \div\}$, and

- Conditional statements of the form "if $v \geq 0 \dots$ else $\dots$".

In both cases, $v$ and $v'$ are either inputs or values previously computed by the algorithm.

The intermediate values $v, v', v''$ computed by the GJ algorithm $\Gamma$ can be considered as rational functions of its real-valued inputs $\alpha$. Based on its intermediate values, one can define the *degree* and the *predicate complexity*, which serve as complexity measures for the GJ algorithm.

**Definition A.2** (Complexities of GJ algorithm, (Bartlett et al., 2022)). The degree of a GJ algorithm is the maximum degree of any rational function that it computes of the inputs. The predicate complexity of a GJ algorithm is the number of distinct rational functions that appear in its conditional statements. Here, the degree of rational function $f(\alpha) = \frac{g(\alpha)}{h(\alpha)}$, where $g$ and $h$ are two polynomials in $\alpha$, is $\deg(f) = \max\{\deg(g), \deg(h)\}$.

For a parameterized function class $\mathcal{L}$ of which each function can be described by a GJ algorithm with bounded complexities, the following result establishes a concrete upper-bound for the pseudo-dimension $\text{Pdim}(\mathcal{L})$.

**Theorem A.3** (Bartlett et al. (2022, Theorem 3.3)). *Suppose that each function $\ell_\alpha \in \mathcal{L}$ is specified by $p$ real parameters $\alpha \in \mathbb{R}^p$. Suppose that for every problem instance $x \in \mathcal{X}$ and real-valued threshold $t \in \mathbb{R}$, there is a GJ algorithm $\Gamma_{x,t}$ that, given $\ell_\alpha \in \mathcal{L}$, returns "true" if $\ell_\alpha(x) \geq t$ and "false" otherwise. Assume that $\Gamma_{x,t}$ has degree $\Delta$ and predicate complexity $\Lambda$. Then, $\text{Pdim}(\mathcal{L}) = \mathcal{O}(p \log(\Delta\Lambda))$.*

The GJ algorithm $\Gamma_{x,t}$ in Theorem A.3 is determined for each specific problem instance $x$ and a real-valued threshold $t$. The input of $\Gamma_{x,t}$ is the hyperparameter $\alpha$ that parameterizes $\ell_\alpha$.

# B. Proofs for Section 4

In this section, we will present the formal proof for Theorem 4.1.

*Proof of Theorem 4.1.* By Theorem 2.5, there exists an equivalent QFF $\Psi_{x,t}(\alpha)$ such that $\Phi_{x,t}(\alpha) \Leftrightarrow \Psi_{x,t}(\alpha)$. Moreover, $\Psi_{x,t}(\alpha)$ is a Boolean combination of $I$ atomic polynomial predicates in $\alpha$, with degree at most $\Delta_{QE}$, where

$$I \leq M^{\prod_{k=1}^{M}(d_k+1)} \cdot \Delta_f^{\mathcal{O}(p) \prod_{k=1}^{M} d_k},$$

$$\Delta_{QE} \leq \Delta^{\mathcal{O}(\prod_{k=1}^{M} d_k)}.$$

We can construct a GJ algorithm $\Gamma_{x,t}$ to evaluate the formula $\Psi_{x,t}(\alpha)$ as follows:

1. For each polynomial $P_j(\alpha)$ appears in $\Psi_{x,t}(\alpha)$, the algorithm $\Gamma_{x,t}(\alpha)$ computes its intermediate value $v_j = P_j(\alpha)$ using standard operators $(+, -, \times)$. Since $P_j$ are polynomials in $\alpha$, this step is valid.

2. For each predicate $P_j(\alpha)\chi_j 0$, the algorithm checks the condition (e.g., "if $v_j \geq 0$") using a conditional statement.

3. Finally, the algorithm $\Gamma_{x,t}(\alpha)$ combines the Boolean results of these checks according to the AND/OR structure of $\Psi_{x,t}$ to return the final truth.

The degree of $\Gamma_{x,t}$ is simply the maximum degree of the intermediate polynomials computed, which is at most $\Delta_{QE}$. The predicate complexity is the number of distinct polynomials in the conditional statements, which is at most $I$.

Finally, applying Theorem A.3, the pseudo-dimension of $\mathcal{L}$ is upper-bounded by:

$$\mathcal{O}(p \log(I \cdot \Delta_{QE})) = \mathcal{O}\left(p \log M^{\prod_{k=1}^{M}(d_k+1)} \cdot \Delta^{2\mathcal{O}(p)\prod_{k=1}^{M} d_k}\right)$$
$$= O\left(p \prod_{k=1}^{M}(d_k+1) \log M + p^2 \prod_{k=1}^{M} d_k \log \Delta\right)$$

as desired. $\qquad\square$

## C. Semi-algebraic Functions and Their Properties

We start with the notion of semi-algebraic functions and sets:

**Definition C.1** (Semi-algebraic sets and functions). A subset $A$ of $\mathbb{R}^n$ is called semi-algebraic if it can be described by a finite number of polynomial equalities and inequalities, i.e.,

$$A = \bigcup_{i \in \mathcal{I}}\{x \mid P_i(x) = 0 \text{ and } Q_{i,j}(x) > 0, \forall j \in \mathcal{J}_i\},$$

where $\mathcal{I}$ and $\mathcal{J}_i, i \in \mathcal{I}$ are finite index sets. A function is semi-algebraic if and only if its graph is semi-algebraic.

Semi-algebraic functions are stable under many operations.

**Proposition C.2** (Properties of semi-algebraic functions (Basu et al., 2006, Theorem 2.84 and 2.85)). *The set of semi-algebraic functions is closed under composition, summation, and multiplication.*

It is noteworthy that the set of piecewise polynomial functions (cf. Definition 3.1) is a strict subset of the set of semi-algebraic functions. Indeed, to show that a piecewise polynomial function $g$ is semi-algebraic, it is sufficient to express its graph as:

$$\texttt{graph}\, g = \bigcup_{\sigma \in \Sigma_{f_x}} \left\{(x,y) \mid \left(\bigcap_{k=1}^{M_f} \text{sign}(h(x)) = \sigma_k\right) \cap (y = f_\sigma(x))\right\}.$$

Other examples of semi-algebraic functions in the learning context that are not piecewise polynomial are:

1. Norm $\ell_p, p \in \mathbb{N}$: because its graph is given by:

$$\{(x,y) \mid y > 0, \left(\sum_{i=1}^{d} x_i^p\right) - y^p = 0\} \cup \{(0,0)\} \subseteq \mathbb{R}^{d+1}.$$

Note that $\ell_p$ is not piecewise polynomial because it is equal to $(\sum_{i=1}^{p} x_i^p)^{\frac{1}{p}}$.

2. The $\ell_2/\ell_1$ ratio, i.e., $\|x\|_2/\|x\|_1$: Note that $f(x,y) = x/y$ is semi-algebraic since their graph is $\{(x,y) \mid xy = 1\}$. The $\ell_2/\ell_1$ ratio is, thus, also semi-algebraic because it is the composition of $g$ and $f$, where:

$$g : \mathbb{R}^d \to \mathbb{R}^2, \qquad x \mapsto \begin{pmatrix} \|x\|_2 \\ \|x\|_1 \end{pmatrix}.$$

The $\ell_2/\ell_1$ ratio is not piecewise polynomial either because it is a rational function (and not polynomial).

3. Group LASSO: Let $(\theta_1, \ldots, \theta_p) \in \mathbb{R}^{d_1} \times \ldots \times \mathbb{R}^{d_p}$ be a decomposition of $\theta \in \mathbb{R}^d$, then the group LASSO is given by:

$$f(\theta) = \sum_{i=1}^{p} \|\theta_p\|_2.$$

As seen previously, $\|\cdot\|_2$ is semi-algebraic, and semi-algebraic functions are stable under summation; group LASSO is also semi-algebraic. It is not piecewise polynomial either, since it equals the sum of $\ell_2$ norms.

# D. Proofs of Section 5

## D.1. Upper bound

*Proof of Theorem 5.1.* To apply Theorem 4.1, given a problem instance $x$ and a real-valued threshold $t$, our goal is to construct a polynomial FOL formula $\Phi_{x,t}(\alpha)$ equivalent to $\mathbb{I}(\ell_\alpha(x) \geq t)$. Since $\ell_\alpha(x) = \min_{\theta \in \Theta} f_x(\alpha, \theta)$ is a minimization over $\Theta$, the condition $\ell_\alpha(x) \geq t$ is equivalent to stating that for all parameter $\theta \in \Theta$, the function value $f_x(\alpha, \theta)$ is greater or equal than $t$, and $\Phi_{x,t}(\alpha)$ is defined as

$$\Phi_{x,t}(\alpha) \triangleq (\forall \theta \in \mathbb{R}^d)[(\theta \in \Theta) \Rightarrow f_x(\alpha, \theta) \geq t]$$
$$= (\forall \theta \in \mathbb{R}^d)[\neg(\theta \in \Theta) \vee (f_x(\alpha, \theta) \geq t)].$$

Here, we use the logical identity $(A \Rightarrow B) = (\neg A \vee B)$ (i.e., not $A$ or $B$). The task now is to analyze the structural complexity of $\Phi_{x,t}(\alpha)$:

- The formula involves exactly one block of quantifiers: $(\forall \theta \in \mathbb{R}^p)$. Thus, the number of quantifier alternations is $K = 1$, and the dimension of the quantified variables is $d_1 = d$.

- The formula involves two types of predicates:
    - Domain constraints: because $\Theta = [\theta_{\min}, \theta_{\max}]^d$ is a box in $\mathbb{R}^d$, cheking the condition $\theta \in \Theta$ requires evaluating $2d$ linear inequalities (i.e., $\theta_j \geq \theta_{\min}$ and $\theta_j \leq \theta_{\max}$, for $j = 1, \ldots, d$).
    - Function structure: The condition $f_x(\alpha, \theta) \geq t$ relies on the piecewise polynomial structure of $f_x$ (cf. Definition 3.1). Formally, this condition holds if the pair $(\alpha, \theta)$ falls into a specific region indexed by a binary vector $\sigma = (\sigma_1, \ldots, \sigma_{M_f}) \in \Sigma_{f_x}$, and the corresponding value polynomial $P_{x,\sigma}(\alpha, \theta)$ that $f_x$ admits in such region satisfies $P_{x,\sigma}(\alpha, \theta) \geq t$. We can express this logically as disjunctions over all valid sign patterns $\Sigma_f$:

$$\bigvee_{\sigma \in \Sigma_f} \Big( \underbrace{\Big[ \bigwedge_{j=1}^{M_f} \mathrm{sign}(h_{x,j}(\alpha, \theta)) = \sigma_j \Big]}_{\text{Region Check}} \wedge \underbrace{[P_{x,\sigma}(\alpha, \theta) \geq t]}_{\text{Value Check}} \Big),$$

where $h_{x,j}$ and $P_{x,\sigma}$ are boundary and piece polynomials in the piecewise polynomial structure of $f_x$.

Consequently, the set of atomic polynomials appearing in this formula consists of: (1) $M_f$ boundary polynomials $\{h_{x,1}, \ldots, h_{x,M_f}\}$, (2) at most $T_f$ polynomial $\{P_{x,\sigma} - t\}_{\sigma \in \Sigma_{f_x}}$.

Therefore, the total number of distinct atomic predicates is bounded by $M_{\text{total}} = M_f + T_f + 2d$, and the maximum degree is $\Delta_f$ (as linear constraints have degree 1). □

## D.2. Lower-bound

In this section, we will show that the factor $pd \log \Delta_f \, \mathrm{Pdim}(\mathcal{L}) = \Omega(pd \log \Delta_f)$ is unavoidable. Consequently, in many cases, if we treat the number of hyperparameters $p$ as a small constant, then the factor $\Theta(d \log \Delta_f)$ in Theorem 5.1 is *tight*. The lower-bound construction is inspired by the *bit-extraction technique*, but requires a non-trivial stabilization argument to handle the implicit optimization problem in the definition of the loss function class. To begin with, we recall a standard property of the *coercive function*, which is helpful for the proof of the lower bound.

**Lemma D.1** (Extreme value theorem for coercive function). *If $P(\theta)$ is a continuous, coercive function $(P(\theta) \to \infty$ as $\|\theta\|_2 \to \infty)$ on an unbounded, closed set, then $P(\theta)$ attains global minimum.*

**Theorem 5.2 (restated).** *Let $\mathcal{A} = \mathbb{R}^p$, $\Theta = \mathbb{R}^d$. Then for every sufficiently large $\Delta_f > 0$, there exists a function class $\mathcal{L} = \{\ell_\alpha : \mathcal{X} \to \mathbb{R} \mid \alpha \in \mathcal{A}\}$, where $\ell_\alpha(x) = \min_{\theta \in \Theta} f(x, \alpha, \theta)$, and $f(x, \alpha, \theta)$ is a degree at most $\Delta_f$ for any problem instance $x \in \mathcal{X}$, such that $\mathrm{Pdim}(\mathcal{L}) = \Omega(pd \log \Delta_f)$.*

*Proof of Theorem 5.2.* By definition, to show $\mathrm{Pdim}(\mathcal{L}) = \Omega(pd \log \Delta_f)$, we have to show that there exists $N = \Omega(pd \log \Delta_f)$ problem instances $x_1, \ldots, x_N \in \mathcal{X}$ and $N$ real-valued threshold $\tau_1, \ldots, \tau_N \in \mathbb{R}$ such that for any bit vector $y \in \{0,1\}^N$, there exists a hyperparameter $\alpha_y \in \mathcal{A}$ such that

$$\mathbb{I}(\ell_{\alpha_y}(x_t) - \tau_t \geq 0) = y_t, \quad t = 1, \ldots, N.$$

In other words, the function class $\mathcal{L}$ parameterized by $\mathcal{A}$ can *shatter* the set of problem instances $\{x_1, \ldots, x_N\}$, with $\tau_1, \ldots, \tau_n$ *witnesses the shattering*. In the following, we will first construct the set of problem instances $\{x_1, \ldots, x_N\}$, where $N = \Omega(pd \log \Delta_f)$, and then the objective $f(x, \alpha, \theta)$ which defines the function class $\mathcal{L}$.

**The construction of problem instances $x_t$.** Let $K = \lfloor \Delta_f / 2 \rfloor$, $B = \lfloor \log_2 K \rfloor$. Let $N = p \cdot d \cdot B$, then it is obvious that $N = \Omega(pd \log \Delta_f)$. For the triplet $(j, i, b)$, where

- $j \in \{1, \ldots, p\}$ specifies the dimensionality of the parameter $\alpha = (\alpha_j)_{j=1}^p \in \mathcal{A} \subset \mathbb{R}^p$,

- $i \in \{1, \ldots, d\}$ specifies the target base-$K$ digit of $\alpha_j$ (and the dimension of $\theta$), and

- $b \in \{1, \ldots, B\}$ specifies the exact bit location to extract,

we define the problem instance $x^{(j,i,b)}$ as the tuple of *one-hot* vectors $(u, v, w) \in \{0,1\}^p \times \{0,1\}^d \times \{0,1\}^B = \mathcal{X}$. Here, $u_j = 1, v_i = 1, w_b = 1$, and all other entries are 0; that is, for the problem instance $x^{(j,i,b)} = (u, v, w)$, we have

$$u = (0, \ldots, 0, u_j = 1, 0 \ldots 0) \in \{0,1\}^p,$$
$$v = (0, \ldots, 0, v_i = 1, 0 \ldots 0) \in \{0,1\}^d,$$
$$w = (0, \ldots, 0, w_b = 1, 0 \ldots 0) \in \{0,1\}^B.$$

**The construction of the objective $f(x, \alpha, \theta)$.** We define the training objective $f(x, \alpha, \theta)$ as follow:

$$f(x, \alpha, \theta) = C \underbrace{\sum_{m=1}^{d} \prod_{k=0}^{K-1} (\theta_m - k)^2}_{\text{1. Grid Penalty } C \cdot P_{grid}(\theta)} + \underbrace{\left( \sum_{n=1}^{p} u_n \alpha_n - \sum_{m=1}^{d} \theta_m K^{m-1} \right)^2}_{\text{2. Selector Penalty}} + \underbrace{0.5 \sum_{m=1}^{d} \sum_{c=1}^{B} v_m w_c E_c(\theta_m)}_{\text{3. Bit Extractor}}.$$

Here, $C > 0$ is a sufficiently large positive constant (independent of the choice of binary vector $y$ as well as the index $y, j, i, b$), and $E_c(t)$, where $t \in \mathbb{R}$ is a bit-extraction polynomial

$$E_c(t) = \sum_{j=0}^{K-1} \beta_{j,c} \left( \prod_{\substack{m=0 \\ m \neq j}}^{K-1} \frac{t - m}{j - m} \right),$$

and $\beta_{j,c} \in \{0,1\}$ denotes the $c^{th}$-bit of the integer $j$ in its binary representation. Specifically, if $t \in \{0, 1, \ldots, K-1\}$, then $E_c(t)$ is the $c^{th}$ bit of $t$ in the binary form. We then define $\ell_\alpha(x) = \min_{\theta \in \mathbb{R}^d} f(x, \alpha, \theta)$, and $\mathcal{L} = \{\ell_\alpha : \mathcal{X} \to \mathbb{R} \mid \alpha \in \mathbb{R}^p\}$. We will elaborate on the meaning of each term *Grid Penalty, Selector Penalty, and Bit Extractor* in the proof of the following claims.

We consider an additional function $\ell_\alpha^{\mathcal{K}}(x) = \min_{\theta \in \mathcal{K}^d} f(x, \alpha, \theta)$, where $\mathcal{K} = \{0, \ldots, K-1\}$, is the value function restricted on the parameter grid $\mathcal{K} \subsetneq \Theta$ instead of the whole parameter domain $\Theta$.

We will now claim that: (1) for any $y$, we can construct $\alpha_y$ such that $\ell_{\alpha_y}^{\mathcal{K}}(x^{(j,i,b)}) = \frac{y^{(j,i,b)}}{2}$, and (2) we can pick $C$ large enough such that $\ell_{\alpha_y}(x^{(j,i,b)}) \in (\ell_{\alpha_y}^{\mathcal{K}}(x^{(j,i,b)}) - 0.1, \ell_{\alpha_y}^{\mathcal{K}}(x^{(j,i,b)}))$.

**Claim 1:** $\ell_{\alpha_y}^{\mathcal{K}}(x^{(j,i,b)}) = \frac{y^{(j,i,b)}}{2}$. For any $y \in \{0,1\}^{p \times d \times B}$, let $\alpha_y = (\alpha_1, \ldots, \alpha_p)$ as follows

$$\alpha_j = \sum_{i=1}^{d} \underbrace{\left( \sum_{b=1}^{B} y_{j,i,b} 2^{b-1} \right)}_{D_{j,i}} K^{i-1}, \quad j = 1, \ldots, p. \tag{6}$$

Here, we can understand the term $D_{j,i} \in \{0, \ldots, K-1\}$ as the decimal form of the binary vector $y_{j,i}$ (uniquely encodes binary vector $(y_{j,i,1}, \ldots, y_{j,i,B})$). Therefore, $\alpha_j$ can be understood as an integer whose base-$K$ digits are exactly

$D_{j,1}, \ldots, D_{j,d}$, and it is a unique way to encode all the bits of the binary matrix $y_j$, for $j = 1, \ldots, p$, onto a single decimal value $\alpha_j$.

Using such construction of $\alpha_y$, for any input problem instance $x^{(j,i,b)}$, the function $f(x, \alpha, \theta)$ becomes

$$f(x^{(j,i,b)}, \alpha_y, \theta) = C \cdot P_{grid}(\theta) + \left( \sum_{m=1}^{d} \theta_m K^{m-1} - \alpha_j \right)^2 + \frac{1}{2} E_b(\theta_i).$$

We have the following observations:

- **The perfect key**: at $\theta^* = (D_{j,1}, \ldots D_{j,d}) \in \mathcal{K}^d$, the grid penalty $C \cdot P_{grid}(\theta) = 0$, the selection penalty $\left( \sum_{m=1}^{d} \theta_m K^{m-1} - \alpha_j \right)^2 = 0$. By the construction, $f(x^{(j,i,b)}, \alpha_y, \theta) = \frac{1}{2} E_b(D_{j,i})$, which is exactly the $b^{th}$ bit of $D_{j,i} = y_{j,i,b}$. So $f(x^{(j,i,b)}, \alpha_y, \theta^*) = \frac{y_{j,i,b}}{2}$.

- **Every other key is wrong**: At $\theta \in \mathcal{K}^d \setminus \{\theta^*\}$, the grid penalty is $C \cdot P_{grid}(\theta) = 0$, the selector penalty $\left( \sum_{m=1}^{d} \theta_m K^{m-1} - \alpha_j \right)^2 \geq 1$, and the bit extractor term $\frac{1}{2} E_b(\theta_i) \geq 0$. This means $f(x^{(j,i,b)}, \alpha_y, \theta^*) \leq f(x^{(j,i,b)}, \alpha_y, \theta)$.

Hence, we claim that $\ell_{\alpha_y}^{\mathcal{K}}(x^{(j,i,b)}) = \frac{y^{(j,i,b)}}{2}$.

**Claim 2: For $C$ large enough, $\ell_{\alpha_y}(x^{(j,i,b)}) \in (\ell_{\alpha_y}^{\mathcal{K}}(x^{(j,i,b)}) - 0.1, \ell_{\alpha_y}^{\mathcal{K}}(x^{(j,i,b)}))$.** When $y, x^{(j,i,b)}$, and $\alpha_y$ are fixed and defined as above, we abuse notation and let $f(\theta) \triangleq f(x^{(j,i,b)}, \alpha_y, \theta) = C \cdot P(\theta) + \psi_y^{(j,i,b)}(\theta)$, where

$$P(\theta) = P_{grid}(\theta), \quad \psi_y^{(j,i,b)}(\theta) = \left( \sum_{m=1}^{d} \theta_m K^{m-1} - \alpha_j \right)^2 + \frac{1}{2} E_b(\theta_i).$$

**The upper bound:** At $\theta^*$, $P(\theta^*) = 0$, and therefore

$$\ell_{\alpha_y}(x^{(j,i,b)}) = f(\theta_{cont}) \leq \psi_y^{(j,i,b)}(\theta^*) = \ell_{\alpha_y}^{\mathcal{K}}(x^{(j,i,b)}),$$

where $\theta_{cont} \in \arg\min_{\theta \in \mathbb{R}^d} f(\theta)$.

Next, we choose the value $C$ properly so that it could be independent of the binary vector $y$ and the indexes $(j, i, b)$.

**The lower bound:** Note that $\psi_y^{(j,i,b)}(\theta)$ is a polynomial of $\theta$, then its derivative is bounded in $[-0.5, K]^d \supset \mathcal{K}^d$. Therefore, it is $L$-Lipschitz continuous in $[-0.5, K]^d$, for some $L = L(j, i, b, y) > 0$. Denote $\delta = \min\{0.25, \frac{0.1}{L}\}$, we claim that any $\theta_{cont}$ has to be close to $\theta^*$, i.e., $\|\theta^* - \theta_{cont}\|_2 < \delta$. Assume that this is not the case, then $\theta_{cont}$ must fall into one of the following categories: (1) there exists $v \in \mathcal{K}^d \setminus \theta^*$ such that $\|\theta_{cont} - v\|_2 < \delta$, and (2) there does not exist $v \in \mathcal{K}^d$ such that $\|\theta_{cont} - v\|_2 < \delta$, i.e., $\theta \in \mathbb{R}^d \setminus \cup_{v \in \mathcal{K}^d} \mathcal{B}(v, \delta)$.

For Case 1, since $\psi_y^{(j,i,b)}(\theta)$ is $L$-Lipschitz in $\mathcal{B}(v, \delta) \subset [-0.5, K]^d$, we have

$$\psi_y^{(j,i,b)}(\theta) \geq \psi_y^{(j,i,b)}(v) - L\delta \geq \psi_y^{(j,i,b)}(\theta^*) + 0.5 - L\delta \geq \psi_y^{(j,i,b)}(\theta^*) + 0.4.$$

This means that $f(\theta) = C \cdot P(\theta) + \psi_y^{(j,i,b)}(\theta) \geq \psi_y^{(j,i,b)}(\theta) \geq \psi_y^{(j,i,b)}(\theta^*) + 0.4$, which is a contradiction.

For Case 2, since $P(\theta)$ is a continuous, coercive function, and $\mathcal{R} = \mathbb{R}^d \setminus \cup_{v \in \mathcal{K}^d} \mathcal{B}(v, \delta)$ is a closed, unbounded set, from the Extreme value theorem (Lemma D.1), there exists $\bar{\theta}_y^{(j,i,b)} \in \mathcal{R}$ such that $P(\bar{\theta}_y^{(j,i,b)}) = \min_{\theta \in \mathcal{R}} P(\theta)$. We then define $\mu = \mu_y^{(j,i,b)} = P(\bar{\theta}_y^{(j,i,b)})$. Note that $\mu > 0$ due to the squared form of $P(\theta)$, and it cannot achieve 0 as $\nu \notin \mathcal{K}^d$. Now note that: (1) $P(\theta)$ is a polynomial of degree $2K$, (2) $\psi_y^{(j,i,b)}(\theta)$ is a polynomial of degree at most $K$. Therefore, there exists some sufficiently large threshold $T > 0$ such that for any $\theta$ such that $\|\theta\|_\infty \geq T$, we have $P(\theta) + \psi_y^{(j,i,b)}(\theta) \geq 1$. This will lead to the following cases:

- If $\theta_{cont} \in \mathcal{R} \cap \{\theta : \|\theta\|_\infty \geq T\}$: then $f(\theta_{cont}) = C \cdot P(\theta_{cont}) + \psi_y^{(j,i,b)}(\theta_{cont}) \geq 1 > f(\theta^*)$ which is a contradiction.

- If $\theta_{cont} \in \mathcal{R} \setminus \{\theta : \|\theta\|_\infty \geq T\}$: Since $\mathcal{R} \setminus \{\theta : \|\theta\|_\infty \geq T\} \subset \{\theta : \|\theta\|_\infty < T\}$ is bounded, and $\psi_y^{(j,i,b)}(\theta)$ is a polynomial, there exists $M = M_y^{(j,i,b)} > 0$ such that $\psi_y^{(j,i,b)}(\theta) \geq -M$ for $\theta \in \mathcal{R} \setminus \{\theta : \|\theta\|_\infty \geq T\}$. We then simply choose $C > \max_{j,i,b,y} \left( 1 + \frac{\psi_y^{(j,i,b)}(\theta^*) + M_y^{(j,i,b)}}{\mu_y^{(j,i,b)}} \right)$, and therefore

$$f(\theta_{cont}) \geq C\mu - M > \psi_y^{(j,i,b)}(\theta^*) = f(\theta^*),$$

which is a contradiction.

Therefore, it must be the case $\|\theta^* - \theta_{cont}\|_2 \leq \delta$, and therefore

$$\psi_y^{(j,i,b)}(\theta_{cont}) \geq \psi_y^{(j,i,b)}(\theta^*) - L\|\theta_{cont} - \theta^*\| \geq \psi_y^{(j,i,b)}(\theta^*) - L\delta \geq \psi_y^{(j,i,b)}(\theta^*) - 0.1$$

where the final inequality comes from the fact that $\delta \leq \frac{0.1}{L}$. This implies:

$$f(\theta_{cont}) \geq \psi_y^{(j,i,b)}(\theta_{cont}) \geq \psi_y^{(j,i,b)}(\theta^*) - 0.1 = f(\theta^*) - 0.1.$$

From the two claims above, we have

$$\ell_{\alpha_y}(x^{j,i,b}) \in \left[ \frac{y^{(j,i,b)}}{2} - 0.1, \frac{y^{(j,i,b)}}{2} \right].$$

Now, if we choose $\tau^{(j,i,b)} = 0.25$, then: (1) if $y^{(j,i,b)} = 1$, $\ell_{\alpha_y}(x^{j,i,b}) - \tau^{(j,i,b)} > 0$ or $\mathbb{I}(\ell_{\alpha_y}(x^{j,i,b}) - \tau^{(j,i,b)} > 0) = y^{(j,i,b)}$, (2) $y^{(j,i,b)} = 0$, $\ell_{\alpha_y}(x^{j,i,b}) - \tau^{(j,i,b)} < 0$ or $\mathbb{I}(\ell_{\alpha_y}(x^{j,i,b}) - \tau^{(j,i,b)} > 0) = y^{(j,i,b)}$. This concludes the proof. $\square$

## E. Proofs of Section 6

*Proof of Theorem 6.1.* Similar to the proof of Theorem 5.1, the idea is to use Theorem 4.1 by showing that: given a problem instance $x$ and a real-valued threshold $t$, there is a polynomial FOL $\Phi_{x,t}(\alpha)$ equivalent to $\mathbb{I}(\ell_x(\alpha) \geq t)$ with bounded complexities. Such a polynomial FOL can be defined as follows:

$$\Phi_{x,t}(\alpha) \triangleq (\forall \theta \in \mathbb{R}^d) \left[ (\theta \in \mathcal{S}(x, \alpha)) \Rightarrow (g(x, \alpha, \theta) \geq t) \right]$$
$$= (\forall \theta \in \mathbb{R}^d) \left[ \neg(\theta \in \mathcal{S}(x, \alpha)) \vee (g(x, \alpha, \theta) \geq t) \right].$$

Here, we again use the identity $(A \Rightarrow B) = (\neg A \vee B)$. We first expand the optimality constraints $\theta \in \mathcal{S}(x, \alpha)$. Note that a parameter $\theta$ is not optimal (i.e., $\neg(\theta \in \mathcal{S}(x, \alpha))$ if it is not in the region $\Theta$ or if there exists a better candidate $\theta'$ such that $f_x(\alpha, \theta') < f_x(\alpha, \theta)$. Therefore, $\neg(\theta \in \mathcal{S}(x, \alpha))$ can be rewritten as

$$(\theta \notin \Theta) \vee \left[ (\exists \theta' \in \mathbb{R}^d)[(\theta' \in \Theta) \wedge (f_x(\alpha, \theta') < f_x(\alpha, \theta))] \right].$$

Let $L_1 = (\theta' \in \Theta) \wedge (f_x(\alpha, \theta') < f_x(\alpha, \theta))$, which is the logical sentence for *optimality check*, we can then write $\Phi_{x,t}(\alpha)$ as

$$(\forall \theta \in \mathbb{R}^d)(\exists \theta' \in \mathbb{R}^d) \left[ \underbrace{\theta \notin \Theta}_{\text{Domain check}} \vee \underbrace{(g_x(\alpha, \theta) \geq t)}_{\text{Validation check}} \vee L_1 \right].$$

We now analyze the structural complexity of $\Phi_{x,t}(\alpha)$:

- The formula involves two blocks of quantifiers, each with dimension $d$. Therefore, $K = 2$, and the complexity scales with $\prod_{i=1}^{K}(d_k + 1) = \mathcal{O}(d^2)$.

- The atomic polynomial predicates required to express $\Phi_{x,t}(\alpha)$ include three components:

  - Domain constraints ($\theta' \notin \Theta$ and $\theta \in \Theta$): since $\Theta = [\theta_{\min}, \theta_{\max}]^d$, checking those constraints take $\mathcal{O}(d)$ atomic predicates of degree 1.

- Validation check $(g_x(\alpha, \theta) \geq t)$: this relies on the piecewise structure of $g_x$ (Definition 3.1). Concretely, this condition holds if the pair $(\alpha, \theta)$ falls into the specific region indexed by a binary vector $\sigma = (\sigma_1, \ldots, \sigma_{M_g}) \in \Sigma_{g_x}$, and the corresponding value polynomial $P_{g_x, \sigma}$ that $g_x$ admits in such region satisfies $P_{g_x, \sigma}(\alpha, \theta) \geq t$. We can express $(g_x(\alpha, \theta) \geq t)$ logically as:

$$\bigvee_{\sigma \in \Sigma_{g_x}} \left( \left[ \bigwedge_{j=1}^{M_g} \operatorname{sign}(h_{g_x, j}(\alpha, \theta)) = \sigma_j \right] \wedge [P_{g_x, \sigma}(\alpha, \theta) \geq t] \right).$$

- Optimization check $(f_x(\alpha, \theta) < f_x(\alpha, \theta))$: here, we compare the function $f_x$ evaluated at two points $(\alpha, \theta)$ and $(\alpha, \theta')$. To do so, we must determine the active regions (indexed by binary vectors $\sigma, \sigma' \in \Sigma_{f_x}$) for both points simultaneously. The condition can then be expressed as:

$$\bigvee_{\sigma \in \Sigma_{f_x}} \bigvee_{\sigma' \in \Sigma_{f_x}} \left( \operatorname{Where}_{f_x}^{\sigma}(\alpha, \theta) \wedge \operatorname{Where}_{f_x}^{\sigma'}(\alpha, \theta') \wedge L \right),$$

where $L = (P_{f_x, \sigma'}(\alpha, \theta') < P_{f_x, \sigma}(\alpha, \theta))$ is the value comparison term, and $\operatorname{Where}_{f_x}^{\sigma}(\alpha, \theta)$ is a shorthand for the conjunction of signs of $M_f$ boundary polynomials evaluated at $(\alpha, \theta)$. The atomic predicates involved here are: (1) the boundary polynomials of $f_x$ evaluated at $\theta$: $\{h_{f_x, j}(\alpha, \theta)\}_{j=1}^{M_f}$, the boundary polynomials of $f$ evaluated at $\theta'$: $\{h_{f_x, j}(\alpha, \theta')\}_{j=1}^{M_f}$, and the pairwise difference of piece polynomials: $\{P_{f_x, \sigma}(\alpha, \theta) - P_{f_x, \sigma'}(\alpha, \theta')\}_{\sigma, \sigma' \in \Sigma_{f_x}}$.

Summing these components, we conclude that the total number of distinct atomic polynomials $M_{\text{total}}$ is bounded by

$$M_{\text{total}} \leq \underbrace{4d}_{\text{Domain}} + \underbrace{(M_g + T_g)}_{\text{Validation}} + \underbrace{(2M_f + T_f^2)}_{\text{Optimization}} = \mathcal{O}(d + M_g + T_g + M_f + T_f^2).$$

The maximum degree $\Delta_{\text{total}}$ is defined by the highest degree among these polynomials, i.e., $\Delta_{\text{total}} = \max(\Delta_f, \Delta_g)$. Substituting this into Theorem 4.1, we have the postulated claim. $\qquad \square$

## F. Proofs and Additional Results for Section 7

### F.1. Proofs

We now present the formal proof of Theorem 7.2.

*Proof of Theorem 7.2.* In this scenario, we can apply the GJ framework directly without invoking quantifier elimination. Besides, since $\theta^*(x, \alpha)$ is now a rational function of $\alpha$, we expect that the final bound should only depend on the dimensionality of $\alpha$ instead of $\theta$. Recall that, to give an upper-bound for the pseudo-dimension of $\mathcal{L}$ using GJ framework, for any problem instance $x \in \mathcal{X}$ and and any real-valued threshold $t \in \mathbb{R}$, we want to show that the computation of $\mathbb{I}(\ell_x^*(\alpha) \geq t)$, where $\ell_x^*(\alpha) \ell_\alpha(x) = k_x(\alpha, \theta^*(x, \alpha))$, can be described by a GJ algorithm (Definition A.1) with bounded complexities.

At the high level idea, the computation of $\mathbb{I}(\ell_x^*(\alpha) \geq t)$ can be divided into the following steps: (1) locating the form of $\theta^*(x, \alpha)$ as a rational function of $\alpha$ based on the relative position of $\alpha$, and (2) locating the polynomial form of the objective $h_x(\alpha, \theta)$, and (3) calculating $\mathbb{I}(\ell_x^*(\alpha) \geq t) = \mathbb{I}(k_x(\alpha, \theta^*(x, \alpha)) \geq t)$ with a GJ algorithm, which is valid since $k_x(\alpha, \theta^*(x, \alpha))$ is now a rational function of $\alpha$.

First, to locate the form of $\theta^*(x, \alpha)$, based on its piecewise rational structure, we first calculate the vector $\sigma(\alpha)$, where $\sigma(\alpha)_i = \operatorname{sign}(h_{x,i}(\alpha))$. This requires at most $M_{\text{path}}$ conditional statements, involving at most $M_{\text{path}}$ distinct rational functions of $\alpha$. Second, for the rational form of $k_x(\alpha, \theta)$, again we leverage its piecewise rational structure, and calculate the vector $\kappa(\alpha, \theta^*(x, \alpha))$, where $\kappa(\alpha, \theta^*(x, \alpha))_i = \operatorname{sign}(h_{x,i,k}(\alpha, \theta^*(x, \alpha)))$. Here $h_{x,i}^k$ is the $i^{th}$ boundary rational functions of the objective function $k$ as in Definition 3.1. This requires at most $M_k \cdot T_{\text{path}}$ conditional statements, where $M_k$ is the number of distinct forms of $h_{x,i,k}$ and $T_{\text{path}}$ is the number of distinct forms of $\theta^*(x, \alpha)$. Finally, after the form of $k_x(\alpha, \theta^*(x, \alpha))$ is determined, which is a rational function of $\alpha$, we can now calculate $\mathbb{I}(\ell_x^*(\alpha) \geq t) = \mathbb{I}(k_x(\alpha, \theta^*(x, \alpha)) \geq t)$. The total distinct rational functions involved appearing in the conditional statements when calculating $\mathbb{I}(\ell_x^*(\alpha) \geq t)$ is at most $M_{\text{total}} = M_{\text{path}} + T_{\text{path}} \cdot M_k + T_{\text{path}} \cdot T_k = M_{\text{path}} + T_{\text{path}}(M_k + T_k)$. Here, the factor $T_{\text{path}} \cdot T_k$ comes from the total forms

that $k_x(\alpha, \theta^*(x, \alpha))$ can take. Finally, the maximum degree of the rational functions appearing in the conditional statements when calculating $\mathbb{I}(k_x(\alpha, \theta^*(x, \alpha)) \geq t)$ is at most $\Delta_{\text{total}} = \Delta_k \cdot \Delta_{\text{path}}$, due to the combination in $k_x(\alpha, \theta^*(x, \alpha))$. Applying Theorem A.3 yields the final result. $\qquad\square$

### F.2. Data-driven ElasticNet

In this section, we will use our general bound in Theorem 7.2 to recover the upper-bound for the problem of data-driven tuning ElasticNet across instances presented by Balcan et al. (2023). Since Balcan et al. (2023) also presented a matching lower-bound, this shows that our general bound is tight for some problem.

**Problem settings.** We first briefly introduce the problem of tuning regularization parameters for ElasticNet, previously considered by Balcan et al. (2023). Concretely, consider a problem instance $x = (A, b, A_{\text{val}}, b_{\text{val}})$, where $(A, b) \in \mathbb{R}^{m \times d} \times \mathbb{R}^m$ representing a training set with $m$ samples and $d$ features, and $(A_{\text{val}}, b_{\text{val}}) \in \mathbb{R}^{m' \times d} \times \mathbb{R}^{m'}$ denotes the validation part of the problem instance $x$, we first consider the ElasticNet estimator $\theta(x, \alpha)$ defined as

$$\theta(x, \alpha) = \arg\min_{\theta \in \mathbb{R}^d} \frac{1}{2m} \|b - A\theta\|_2^2 + \alpha_1 \|\theta\|_1 + \alpha_2 \|\theta\|_2^2.$$

Here, $\alpha = (\alpha_1, \alpha_2) \in \mathbb{R}_{>0}^2$ denote the regularization hyperparameters controlling the magnitude of the $\ell_1$ and $\ell_2$ regularization. Then, the solution $\theta(x, \alpha)$ is evaluated in the validation set $(A_{\text{val}}, b_{\text{val}})$ of the problem instance $x$

$$\ell_\alpha(x) = \frac{1}{2m'} \|b_{\text{val}} - A_{\text{val}}\theta(x, \alpha)\|_2^2.$$

Assuming that there is an application specific problem distribution $\mathcal{D}$ over the space of problem instance $\mathcal{X} = \mathbb{R}^{m \times d} \times \mathbb{R}^m \times \mathbb{R}^{m' \times d} \times \mathbb{R}^{m'}$, our goal is to answer the question of how many problem instances do we need to learn a good hyperparameter $\alpha$ for the problem distribution $\mathcal{D}$. Denote $\mathcal{L} = \{\ell_\alpha : \mathcal{X} \to [-H, H] \mid \alpha \in \mathbb{R}_{>0}^2\}$, the previous question is equivalent to giving an upper-bound for the pseudo-dimension of $\mathcal{L}$.

**Recovering the pseudo-dimension upper-bound.** We now demonstrate how to use our general result to establish an upper bound for $\text{Pdim}(\mathcal{L})$. First, we will invoke the properties of the solution path of the ElasticNet, a rephrasing of the structural result mentioned in Balcan et al. (2023).

**Proposition F.1** (Theorem 3.2, Balcan et al. (2023)). *For any fixed problem instance, the unique mapping $\alpha \to \theta^*(x, \alpha)$ satisfies Assumption 7.1 with*

- *$T_{\text{path}} = \mathcal{O}(3^d)$: the domain $\mathbb{R}_0^2$ is partitioned into disjoint regions corresponding to the sign patterns (e.g., active sets) of the optimal coefficients. The number of regions is bounded by the total number of sign patterns, i.e., $T_{\text{path}} \leq 3^d$.*

- *$M_{\text{path}} = \mathcal{O}(d3^d)$: the boundaries separating these regions are defined by conditions where a coefficient vanishes. The number of such boundary polynomials is bounded by $M_{\text{path}} \leq d3^d$.*

- *$\Delta_{\text{path}} = \mathcal{O}(d)$: inside each region, the solution is given by a rational function (derived using Cramer's Rule on the active linear system). The degree of these rational functions is bounded by $\Delta_{\text{path}} \leq 2d$.*

We are now ready to recover the upper-bound for the pseudo-dimension of $\mathcal{L}$, which then implies the generalization guarantee for data-driven tuning of the regularization hyperparameters for ElasticNet, by combining Proposition F.1 and Theorem 7.2.

**Corollary F.2.** *Let $\mathcal{L}$ be the class of validation loss functions for ElasticNet as defined above. Then $\text{Pdim}(\mathcal{L}) = \mathcal{O}(d)$.*

*Proof.* We first calculate the total complexities $M_{\text{total}}, \Delta_{\text{total}}$. First note that $p = 2$ (as $\alpha \in \mathbb{R}_{>0}^2$), and then we have:

- The piecewise rational structural complexities of the tuning objective $k_x(\alpha, \theta) = \frac{1}{2} \|b_{\text{val}} - A_{\text{val}}\theta\|_2^2$ are $M_k = 0$, $T_k = 1$, and $\Delta_k = 2$.

- Combining with Proposition F.1, the total complexities are
  - $M_{\text{total}} = M_{\text{path}} + T_{\text{path}}(M_k + T_k) \leq (d+1)3^d$,
  - $\Delta_{\text{total}} = \Delta_k \cdot \Delta_{\text{path}} \leq 4d.$

Finally, applying Theorem 7.2 gives us

$$\text{Pdim}(\mathcal{L}) = \mathcal{O}(p \log(M_{\text{total}} \Delta_{\text{total}})) = \mathcal{O}(d).$$

This completes the proof. □

*Remark* F.3. Note that by Balcan et al. (2023, Theorem 3.5), we have $\text{Pdim}(\mathcal{L}) = \Omega(d)$. Therefore, applying Theorem 7.1 successfully recovers a tight bound for the pseudo-dimension of $\mathcal{L}$.

## G. Proofs of Section 8

### G.1. Data-driven Weighted Group Lasso

*Proof of Theorem 8.1.* The proof is similar to the previous. The only difference is how we describe $f$ using polynomials since the function is no longer piecewise polynomial. This is possible by adding extra $\nu_1, \ldots, \nu_p$ scalar variable and write:

$$f(x, \alpha, \theta) = \|A\theta - b\|^2 + \sum_{i=1}^{p} \alpha_i \nu_i$$

with polynomial constraints:

$$\nu_i^2 = \sum_j [\theta_i]_j^2 \quad \text{and} \quad \nu_i \geq 0, \forall i = 1, \ldots, p. \tag{7}$$

The corresponding first-order formula is given by:

$$\forall \theta \in \mathbb{R}^d, \exists (z, \nu^\theta, \nu^z) \in \mathbb{R}^{d+2p}, (T_1 \vee T_2),$$

where $T_1$ and $T_2$ are:

$$T_1 = (\|A\theta - b\|^2 + \sum_i \alpha_i \nu_i^\theta > \|Az - b\|^2 + \sum_i \alpha_i \nu_i^z) \wedge (\text{conditions (7) for } \nu^\theta \text{ and } \nu^z),$$

$$T_2 = \|A'\theta - b\|^2 \geq t.$$

Overall, we have $\Delta = 2$ and $M = 2(1 + 2p)$. Applying Theorem 4.1, we obtain:

$$\text{Pdim}(\mathcal{L}) = \mathcal{O}(p(d+1)(d+2p+1)\log(2+4p) + p^2(d+1)(d+2p+1)\log 2) = \mathcal{O}(p^3 d + p^2 d^2). \quad \square$$

### G.2. Data-driven Weighted Fused Lasso

We remind the problem setting and prove Theorem 8.2.

To prove the results of the Weighted Fused LASSO, we first need to reformulate the optimization problem into a canonical form. Let $D \in \mathbb{R}^{(d-1) \times d}$ be the first-difference matrix, i.e.,

$$D = \begin{bmatrix} -1 & 1 & 0 & \cdots & 0 & 0 \\ 0 & -1 & 1 & \cdots & 0 & 0 \\ \vdots & \vdots & \vdots & \ddots & \vdots & \vdots \\ 0 & 0 & 0 & \cdots & -1 & 1 \end{bmatrix}.$$

The Weighted Fused LASSO estimation problem can be rewritten as

$$\min_{\theta \in \mathbb{R}^d} \frac{1}{2} \|b - A\theta\|_2^2 + \sum_{i=1}^{p} \alpha_i |(D\theta)_i|,$$

or equivalently,

$$\min_{\theta \in \mathbb{R}^d} \frac{1}{2} \|b - A\theta\|_2^2 + \sum_{i=1}^{p} \alpha_i |z_i|, \text{ s.t. } z = D\theta.$$

First, we will show that the dual form of Weighted Fused LASSO takes the *multi-parametric Quadratic Programming* (mp-QP) (Bemporad et al., 2002).

**Proposition G.1.** *The dual formulation of Weighted Fused LASSO is*

$$\min_{u \in \mathbb{R}^p} \frac{1}{2} \|\tilde{b} - \tilde{A}u\|_2^2, \text{ s.t. } |u_i| \le \alpha_i, i = 1, \dots, p,$$

*where $\tilde{A} = (A^\top A)^{-1/2} D^\top$ and $\tilde{y} = (A^\top A)^{-1/2} A^\top y$.*

*Proof.* We first define the Lagrangian function $L(\theta, z, u)$ as

$$L(\theta, z, u) = \frac{1}{2} \|b - A\theta\|_2^2 + \sum_{i=1}^p \alpha_i |z_i| + u^\top (D\theta - z),$$

where $u \in \mathbb{R}^p$ is a Lagrangian variables. Strong duality holds due to the convexity and linear-quadratic structure of the original problem. We now derive the dual objective function, which is the separable infimum of the Lagrangian function w.r.t. the primal variables $\theta$ and $z$, i.e.,

$$
\begin{aligned}
g(u) &= \inf_{\theta, z} L(\theta, z, u) \\
&= \underbrace{\inf_{\theta} \left( \frac{1}{2} \|y - A\theta\|_2^2 + u^\top D\theta \right)}_{\text{First term}} + \underbrace{\inf_{z} \left( \sum_{i=1}^p (\alpha_i |z_i| - u_i z_i) \right)}_{\text{Second term}}.
\end{aligned}
$$

For the first term, we simply take the gradient w.r.t. $\theta$ and set it to zero, i.e.

$$\nabla_\theta \left( \frac{1}{2} \|b - A\theta\|_2^2 + u^\top D\theta \right) = -A^\top (b - A\theta) + D^\top u = 0.$$

If $A$ has full column rank (or in the *general position*) (Tibshirani, 2011), we conclude that the optimal $\theta^*(u)$ is

$$\theta^*(u) = (A^\top A)^{-1} (A^\top b - D^\top u).$$

For the second term, the infimum of $\alpha_i |z_i| - u_i z_i$ is bounded only if $|u_i| \le \alpha_i$. If this holds, the infimum is 0; otherwise, it is $-\infty$. This gives us the box constraints $|u_i| \le \alpha_i$. Therefore, we conclude that the dual problem is

$$
\begin{aligned}
\min_{u \in \mathbb{R}^p} \quad & \frac{1}{2} (A^\top y - D^\top u)^\top (A^\top A)^{-1} (A^\top b - D^\top u), \\
\text{s.t.} \quad & |u_i| \le \alpha, i = 1, \dots, p.
\end{aligned}
$$

Simply setting $\tilde{A} = (A^\top A)^{-1/2} D^\top$ and $\tilde{b} = (A^\top A)^{-1/2} A^\top y$, we have the final conclusion. $\square$

Since the above take the form of mp-QP with linear constraints (i.e., $-\alpha \le u_i \le \alpha$), from Theorem 2 in Bemporad et al. (2002), the solution $u^*(x, \alpha)$ is a *piecewise affine* function of $\alpha$, where each piece corresponds to an active constraint. Using this property, we are now ready to prove the generalization guarantee for the problem of data-driven tuning regularization parameters for Weighted Fused LASSO in Theorem 8.2.

*Proof of Theorem 8.2.* To bound the pseudo-dimension, we use our proposed Theorem 7.2. The complexity depends on the number of region $M_{\text{path}}$ in the partition:

- Each critical region corresponds to a stable set of active constraints.

- For each of $p = d - 1$ dual variables, the box constraints allow for at most 3 states at optimality: (1) $u_i = \alpha_i$, (2) $u_i = -\alpha_i$, or (3) $-\alpha_i < u_i < \alpha_i$.

- This means that the number of distinct regions is bounded by $M_{\text{path}} \le 3^p = 3^{d-1}$.

Besides, $\Delta_{\text{path}} = 1$ (affine) and $\Delta_k = 2$ (quadratic of the validation $k(x, \alpha, \theta) = \frac{1}{2} \|b' - A'\theta\|_2^2$). Substituting into Theorem 7.2, we have the final conclusion. $\square$