# OpenReview forum: "Provably Data-driven Multiple Hyper-parameter Tuning with Structured Loss Function"
_ICML.cc/2026/Conference — ICML 2026 regular_

### Official Review · Reviewer_iBHE · 2026-03-08

**Soundness:** 3
**Presentation:** 2
**Significance:** 3
**Originality:** 3
**Overall Recommendation:** 4
**Confidence:** 2

**Summary:**

This paper establishes general learning guarantees for data-driven multi-dimensional hyperparameter tuning. By leveraging tools from statistical learning theory and real algebraic geometry, the authors provide the first general sample complexity guarantees for this setting. They further refine these error bounds by exploiting additional algebraic structure and solution paths.

**Compliance With Llm Reviewing Policy:**

Affirmed.

**Final Justification:**

I thank the authors for providing the detailed rebuttal and discussion. Given the contribution,  I keep my positive score.

**Key Questions For Authors:**

1. To make the paper more self-contained and clarify the contribution, could the authors explain more precisely why the analysis in prior work (e.g., Balcan et al., 2025) is restricted to specific settings?
2. As noted in the conclusion, would it be possible to validate the theoretical results with lower bounds in simple settings or through illustrative numerical experiments?
3. Could the authors provide a concrete example applying the theory beyond the lasso problem? For instance, how might the general guarantees for polynomial first-order logic (FOL) $\Phi$ extend to hyperparameter tuning in neural networks?
4. Bayesian optimization also offers guarantees for hyperparameter tuning under different assumptions. While I understand the settings are not directly comparable, the claim of being the "first general framework" may warrant qualification, and a brief discussion of these alternative guarantees would provide useful context.

Minor:
-	In the last line of Page 4, degrees at most “$\Delta$” should be replaced with $\Delta_f$.

**Limitations:**

yes

**Strengths And Weaknesses:**

Strength:
- The problem addressed is fundamental to learning theory and has broad implications.
- The proposed framework is general, offering guarantees that apply to a wide range of problems.

Weakness:
- Technical density and accessibility: The paper is highly technical and densely packed with theoretical content, introducing numerous definitions that may be unfamiliar to much of the machine learning community. This significantly limits accessibility. A roadmap or diagram illustrating how the various concepts connect in proving the main results would greatly aid readability.
- Suboptimal organization of technical material: The structure of the theoretical exposition could be improved. For instance, Section 3 might be more logically placed before Section 2. Additionally, Section 2.2, which introduces concepts likely unfamiliar even to theory-oriented readers, could be paired more directly with Section 4, as the algebraic structure is not utilized until Theorem 4.1.

---

> ### Author Rebuttal · Authors · 2026-03-31
>
> We thank the reviewer for the positive feedback. We are glad that the reviewers find our contribution fundamental and general. We will address the reviewer's concerns as follows.
>
> ### Questions
> 1. Q: More elaboration on why the analysis in prior work [1] does not work for the general case.
>
>     A. Roughly speaking, the prior work [1] relies on its Lemma 2.1. It says that if the loss function $\ell_x(\alpha) = \inf_{\theta} f(x, \alpha, \theta)$ admits a piecewise continuous form with bounded number of discontinuities $B_1$ points and local extrema $B_2$ points, we can bound the pseudo-dimension of $\mathcal{L}$ based on $B_1$ and $B_2$. However, the discontinuities and local extrema points are properties of a 1-dimensional function ($\alpha \in \mathbb{R}$), there is no clear generalization of this type of result in the high-dimensional case ($\alpha \in \mathbb{R}^p$). Also, their strategy to bound the number of discontinuities and local extrema leverages a terminology called monotonic curve (Definition 19, [1]), a 1-dimensional smooth manifold, leveraging the fact that $\alpha \in \mathbb{R}$. As we can see, their analysis is specifically tailored to the one-dimensional case. We will incorporate this discussion in the revised draft.
>
> 2. Q. Illustration/simple elaboration on the lower-bound
>
>     A: To address this concern, __we provide the following novel lower bound (and its detailed proof) in the revised draft.__
> ----------------------------------------------
> **Theorem**: Let $\mathcal{A} \subset \mathbb{R}^p$, $\Theta = \mathbb{R}^d$. Then, there exists a function class $\mathcal{L} = \{\ell_\alpha: \mathcal{X} \rightarrow \mathbb{R} \mid \alpha \in \mathcal{A}\}$, where $\ell_\alpha(x) = \min_{\theta \in \Theta} f(x, \alpha, \theta)$, and $f(x, \alpha, \theta)$ is a degree at most $\Delta_f$ for any problem instance $x$, such that $\textup{Pdim}(\mathcal{L}) = \Omega(pd\log \Delta_f)$.
>
>
> **See the discussion with Reviewer HBGp for the proof sketch**. This new result shows that **the dependency of $\textup{poly}(pd) \log \Delta_f$ in the upper-bound (Theorem 5.1) is non-negotiable.**
>
> 3. Q: Potential extension to modern (DNNs-related) settings
>
>     A: As discussed with the reviewer rbQS, we note that our results can be extended to other hyperparameter tuning settings in DNNs. In fact, the prior work [1] focused specifically on tuning hyperparameters in DDNs (such as tuning graph kernels in GNNs, etc.). A recent work [2], operated on the same framework, though in a simpler formulation, see e.g., discussion with reviewer rbQS for more details, provided a guarantee for tuning step-size, learning rate schedule, etc.
>
>     Our goal in this work is to answer open questions by [1], but we agree that additional discussion on the applicability of our results to modern settings would help. We will incorporate the changes in the revised draft.
>
> 4. Q: Though not directly comparable, more elaboration on the comparison to Bayesian hyperparameter tuning might help.
>
>     A: We agree that, though not being directly comparable, a more elaborate discussion of the differences between the guarantees of Bayesian optimization and data-driven hyperparameter tuning (this work) should be discussed.
>
>      Roughly speaking, Bayesian optimization operates in a single problem setting, and assumes that the performance of the model w.r.t. hyperparameter can be approximated as a noisy evaluation of a (typically smooth) function, and typically makes assumptions on the form of the noise, the acquisition function, which determines the hyperparameter search space, the prior (e.g., the type of kernel, bandwidth parameter). The guarantee is of the form: for a fixed problem instance/task, if we evaluate the performance of the model w.r.t. different hyperparameters enough times, you can approximate the performance of the model (w.r.t. hyperparameters) well enough, and you can use that to find the optimal hyperparameter for such a fixed problem instance.
>
>     In contrast, the data-driven hyperparameter tuning operates on multiple problem instance settings. In this work, we provide a statistical learning guarantee: assuming that those problem instances/tasks come from the same problem distribution, if we tune the hyperparameter using the observed problem instances, that hyperparameter is guaranteed to have good performance on the new, unseen problem instance from the same problem distribution.  We will incorporate this discussion suggested by the reviewer in the revised draft.
>
> ### Conclusion
> We hope that our answers resolve the reviewer’s concerns, and we are happy to address other concerns if needed. Many thanks!
>
> ### References
> [1] Bacan et al. Sample complexity of data-driven tuning of model hyperparameters in neural networks with structured parameter-dependent dual function, NeurIPS’25
>
> [2] Dravyansh Sharma, Gradient Descent with Provably Tuned Learning-rate Schedules, AISTATS’26

---

> > ### Author Rebuttal · Reviewer_iBHE · 2026-04-01
> >
> > I thank the authors for addressing my questions. My initial evaluation was already positive, and given that this topic is outside my primary area of expertise, I will maintain my positive score.

---

### Official Review · Reviewer_LpHo · 2026-03-09

**Soundness:** 4
**Presentation:** 4
**Significance:** 4
**Originality:** 4
**Overall Recommendation:** 6
**Confidence:** 3

**Summary:**

The paper relies on a standard result in learning theory that states that a
real-valued function class with bounded pseudo-dimension is Probably
Approximately Correct-learnable with empirical risk minimization.
The Authors contribute in several aspects.
First, they consider classes of loss functions described by
a polynomial first-order logic, and prove that the pseudo-dimension is bounded.
Second, they consider loss functions obtained via bi-level
optimization, with a piecewise polynomial structure; they obtain bounds for the
pseudo-dimension. Third, they provide applications in learning problems.

**Compliance With Llm Reviewing Policy:**

Affirmed.

**Final Justification:**

I had no question as i) the paper was (and remains) far from my expertise, ii)
the mathematics are perfectly mastered, and well explained.
By reading the answers to other Reviewers (notably novel bounds, applicability),
my initial gut-feeling is confirmed: the Authors master an original theoretical
question with possible relevant applications. My score was and remains the highest (6).

**Key Questions For Authors:**

I have no question as the paper is much too far from my expertise.
My impression is that the mathematics are perfectly mastered, and well
explained, which can also explain the absence of questions.

**Limitations:**

yes

**Strengths And Weaknesses:**

Soundness
4: excellent

The mathematics seem sound, and perfectly mastered.


Presentation
4: excellent

This paper has a clear progression, which helps the reader grasp the subject,
even if it is tough and well beyond my scope.


Significance
4: excellent

The paper relies on Theorem 2.2 (Pollard, 1984), a standard result in learning
theory that states that a real-valued function class with bounded pseudo-dimension is Probably
Approximately Correct (PAC)-learnable with empirical risk minimization (ERM).
The Authors contribute in several aspects.
First, they consider classes of loss functions described by
a polynomial first-order logic, and prove that the pseudo-dimension is bounded (Theorem 4.1).
Second, they consider loss functions obtained via bi-level
optimization, with a piecewise polynomial structure; they obtain bounds for the
pseudo-dimension (training loss in Theorem 5.1 and validation loss in Theorem 6.1).
Third, they provide applications in learning problems (Section 8).

Even if I am away from my field of expertise, I think that the paper does
address a relevant problem, and I have the feeling that it can advance
understanding and practice in machine learning.


Originality
4: excellent

This is hard to answer as I do not have enough perspective on the subject,
but my gut-feeling suspects that the approach is truly original.

---

> ### Author Rebuttal · Authors · 2026-03-31
>
> We thank the reviewer for very positive feedback. We are glad that the reviewer finds our work has a clear presentation, as well as having a very good understanding of the key contributions that we made. Should the reviewer have any other questions in the rebuttal phase, we are happy to address them. Many thanks!

---

> > ### Author Rebuttal · Reviewer_LpHo · 2026-04-01
> >
> > I had written "I have no question as the paper is much too far from my expertise. My impression is that the mathematics are perfectly mastered, and well explained, which can also explain the absence of questions.". By reading the answers to other Reviewers (notably novel bounds, applicability), my gut-feeling is confirmed: the Authors master an original theoretical question with possible relevant applications. My score was already high; I do not change it.

---

### Official Review · Reviewer_HBGp · 2026-03-13

**Soundness:** 4
**Presentation:** 3
**Significance:** 3
**Originality:** 3
**Overall Recommendation:** 5
**Confidence:** 2

**Summary:**

This paper studies the problem of establishing generalization guarantees for data-driven multi-dimensional hyperparameter tuning when the resulting loss depends on the hyperparameters implicitly through an inner optimization problem (i.e., a bilevel structure). The authors develop a framework that characterizes the complexity of the hyperparameter to loss mapping using pseudo-dimension, which measures the capacity of real-valued function classes and determines how well empirical performance generalizes to new tasks. Their approach shows that when the induced loss functions form a semi-algebraic function class the pseudo-dimension can be bounded using tools from polynomial first-order logic and quantifier elimination in real algebraic geometry. Using this framework, the authors derive generalization guarantees for both training-loss tuning and validation-loss tuning (the bilevel case), and provide refined bounds when explicit solution paths are available. Finally, they apply their theory to obtain guarantees for weighted group LASSO and weighted fused LASSO.

**Compliance With Llm Reviewing Policy:**

Affirmed.

**Final Justification:**

The rebuttal satisfactorily addressed most of my concerns, clarified the applicability of their framework to broader settings, provided additional justification for some of the theoretical results, and discussed extensions. Although its practical impact may be indirect at this stage, I believe it advances the theoretical understanding of an important problem and is likely to stimulate follow-up work. Therefore, I keep recommend this work to be accepted.

**Key Questions For Authors:**

**Questions**

1. Can you provide any evidence, even heuristic or experimental, that the bounds in Theorems 5.1 and 6.1 are not wildly conservative for practical problem sizes?
2. Would fundamentally new ideas be required in order to move away from semi-algebraic functions?
3. Do the authors see any way in which the FOL/quantifier elimination perspective might lead to algorithmic insights or practical tuning procedures?
4. The framework appears to assume that the inner optimization problem is solved exactly. How sensitive are the guarantees to approximate solutions of the inner problem, which is common in practice (e.g., iterative solvers or early stopping)?

**Limitations:**

Yes

**Strengths And Weaknesses:**

**Strengths**

1. The paper is clearly written.
2. The core idea of encoding the loss function’s behavior as a polynomial first-order logic (FOL) formula and applying quantifier elimination is conceptually clean and elegant.
3. The improvement over Goldberg & Jerrum (1993) in Theorem 4.1 looks technically non-trivial and is practically meaningful, as mentioned for e.g. $q \gg p$.
4. The applications to weighted group LASSO and weighted fused LASSO are interesting and demonstrate the practical relevance of the framework.


**Weaknesses**
1. The general bounds (Theorems 5.1 and 6.1) scale polynomially in both $p$ and $d$, whereas the refined bounds under Assumption 7.1 (Theorem 7.2) can scale as $O(d)$ and be tight (as shown in Corollary F.2). However, the tightness of the general bounds remains unclear. The paper acknowledges this limitation, but further discussion/experiments would help clarify how loose these bounds might be in practice.
2. The refined bounds in Theorem 7.2 require a unique, explicitly characterizable piecewise rational solution path. Many practical learning problems do not satisfy this assumption, for example those involving non-convex objectives. The paper would benefit from a clearer discussion of which classes of problems satisfy this assumption and where it is likely to fail.
3. The framework focuses on semi-algebraic function classes, whereas many modern machine learning objectives involve functions such as exponentials, logarithms, or neural network activations that fall outside this class. While the authors mention extending the theory beyond semi-algebraic settings as future work, the current scope may still appear less broad than modern hyperparameter tuning problems.
4. The paper focuses entirely on statistical efficiency (sample complexity) and does not address the computational efficiency of solving the empirical risk minimization problem over multi-dimensional continuous hyperparameter spaces. A brief discussion of whether the FOL/quantifier elimination perspective provides any computational insights or algorithmic implications would strengthen the paper.

---

> ### Author Rebuttal · Authors · 2026-03-31
>
> We thank the reviewer for the positive feedback. We are glad that the reviewers find our theoretical contribution technically non-trivial and practically meaningful. We will address the concerns of the reviewers as follows.
>
> ### Questions
>
> 1. Q: evidence that the upper-bound is not over-conservative:
>
>     A: To address this concern, __we provide the following novel lower bound (and its detailed proof) in the revised draft.__
> ----------------------------------------------
> **Theorem**: Let $\mathcal{A} \subset \mathbb{R}^p$, $\Theta = \mathbb{R}^d$. Then, there exists a function class $\mathcal{L} = \{\ell_\alpha: \mathcal{X} \rightarrow \mathbb{R} \mid \alpha \in \mathcal{A}\}$, where $\ell_\alpha(x) = \min_{\theta \in \Theta} f(x, \alpha, \theta)$, and $f(x, \alpha, \theta)$ is a degree at most $\Delta_f$ for any problem instance $x$, such that $\textup{Pdim}(\mathcal{L}) = \Omega(pd\log \Delta_f)$.
>
> **Proof sketch**. (A) The idea to construct $N = \Omega(pd \log \Delta_f)$ problem instances $x_1, \dots, x_N$, and $N$ rea-valued thresholds $\tau_1, \dots, \tau_N$ sucht ath for any bit vector $y \in \{0, 1\}^N$, there exists $\alpha_y$ such that $\mathbb{I}(\ell_{\alpha_y}(x_i) - \tau_i) \geq 0) = y_i$ for $i = 1, \dots, N$.
>
> Let $K = \lfloor \Delta_f / 2\rfloor$, $B = \lfloor \log_2 K\rfloor$. Let $N = p \cdot d \cdot B$, then it is clear that $N = \Omega(pd \log \Delta_f)$. For each triplet $(j, i, b) \in \{1, \dots, p\} \times \{1, \dots, d\} \times \{1, \dots, B\}$, we define $x^{(j, i, b)}$ as  a tuple of one-hot vectors $(u, v, w) \in \{0, 1\}^p \times \{0, 1\}^d \times \{0, 1\}^B$. Here, $u_j = 1, v_j = 1, w_b = 1$, and all other entries are $0$.
>
> We then define  $f(x, \alpha, \theta)  = C \sum_{m=1}^d \prod_{k=0}^{K-1} (\theta_m - k)^2 + \left( \sum_{m=1}^p u_m \alpha_m - \sum_{m=1}^d \theta_m K^{m-1} \right)^2 + 0.5 \sum_{m=1}^d \sum_{c=1}^B v_m w_c E_c(\theta_m)$, where $E_c(t) = \sum_{j=0}^{K-1} b_{j, c} \left( \prod_{\substack{m=0 \\ m \neq j}}^{K-1} \frac{t - m}{j - m} \right)$ is the bit-extraction polynomial. We then define $g_\alpha(x) = \min_{\theta \in \Theta}f(x, \alpha, \theta)$,
>
> and $g_\alpha(x) =  \min_{\theta \in \\{0, \dots, K - 1\\}^d} f(x, \alpha, \theta)$. Under such construction, we can claim that:
> (1) $g_{\alpha_y}(x^{(i, j, b)}) = \frac{y^{(i, j, b)}}{2}$, and (2) for $C$ large enough, $\ell_{\alpha_y}(x^{(j, i, b)}) \in [g_{\alpha_y}(x^{(j, i, b)}) - 0.1, g_{\alpha_y}(x^{(j, i, b)})]$.
>
> Using the fact above, we can choose $\tau^{(j, i, b)} = 0.25$, and claim that this construction satisfies the statement (A) above, which concludes the proof.
>
> ----------------------------------
> This new result shows that **the dependency of $\textup{poly}(pd) \log \Delta_f$ in the upper-bound (Theorem 5.1) is non-negotiable.**
>
> 2. Q: Is a new idea required for moving away from a semi-algebraic function?
>
>     A: This is a very good catch. As noted in our Conclusion and Future works section, we believe that this approach can be generalized to the structure that is compatible with FOL/QE, for example Pfaffian functions [1] (which includes almost any function used in ML, like $\exp, \log, \tanh$ along with rational functions). We think this is an interesting direction for future work.
>
>
> 3. Q: If FOL/quantifier elimination might lead to algorithm insights or practical tuning problems?
>
>     A: We admit that FOL/QE serve purely as a tool for analyzing the learning-theoretic complexity in this work, and honestly, we are not sure if FOL/QE can bring new algorithmic insights for empirical hyperparameter tuning. But we think this is a very interesting point for future work.
>
> 4. Q: Currently, the inner problem is required to be solved exactly. Can we extend to the case that the inner problem is solved approximately?
>
>     A:  That is a very good catch! Actually, our framework can flexibly handle the case when the inner problem is solved approximately. Roughly speaking, let’s say, in the inner problem $\\min_{\\theta \\in \\Theta} f(x, \\alpha, \\theta)$, we can define the set of $\\epsilon$-approximated solutions  $\\mathcal{S}_\\epsilon(x, \\alpha)$ containing
>
> $\\theta_\\epsilon$ s.t $f(x, \\alpha, \\theta_\\epsilon) + \\epsilon < \\min_{\\theta \\in \\Theta} f(x, \\alpha, \\theta)$.  This is equivalent to the logical sentence $\theta \in \mathcal{S}_\epsilon(x, \alpha) \Leftrightarrow \forall \theta’ \in \Theta, f(x, \alpha, \theta) \leq f(x, \alpha, \theta’) + \epsilon$, which is of our framework.
>
> __We thank the review for the suggestion, and we will incorporate this point in the revised draft.__
>
> ### Conclusion
> We hope that our answers resolve the reviewer’s concerns, and we are happy to address other concerns if needed. Many thanks!
>
> ### References
>
> [1] Balkan et al., Algorithm configuration for structured Pfaffian settings, TMLR’25

---

> > ### Author Rebuttal · Reviewer_HBGp · 2026-04-03
> >
> > I thank the authors for the detailed rebuttal, the responses have clarified my main concerns. I increase my score to 5, supporting acceptance.

---

### Official Review · Reviewer_rbQS · 2026-03-13

**Soundness:** 3
**Presentation:** 4
**Significance:** 3
**Originality:** 3
**Overall Recommendation:** 4
**Confidence:** 3

**Summary:**

This paper studies data-driven hyperparameter tuning in a general bilevel setting. The goal is to show generalization guarantees for the induced class L, especially when the hyperparameter is multi-dimensional. The main technical contribution is a general pseudo-dimension bound derived by expressing threshold predicates in polynomial first-order logic, followed by quantifier elimination and a Goldberg-Jerrum-style complexity argument.

**Compliance With Llm Reviewing Policy:**

Affirmed.

**Final Justification:**

I thank the authors for their rebuttal and agree that the primary focus of this paper is theoretical. Given the novelty and overall quality of the work, I maintain my positive rating and recommend acceptance.

**Key Questions For Authors:**

Could the authors discuss whether the proposed framework can cover more practical modern ML settings, such as learning rate schedule optimization [2] or data reweighting / data mixture optimization [3]? If not, what assumptions of the current theory fail in these cases?

**Limitations:**

Yes.

**Strengths And Weaknesses:**

**Strengths**:
1. The paper makes a meaningful theoretical advance over the prior one-dimensional work of [1]. The earlier paper fundamentally relies on a one-dimensional oscillation/monotonic-curve analysis for scalar hyperparameters. This paper directly addresses that limitation and proposes a substantially different proof route based on logical definability.\
[1] Balcan, Maria-Florina, Anh Tuan Nguyen, and Dravyansh Sharma. "Sample complexity of data-driven tuning of model hyperparameters in neural networks with structured parameter-dependent dual function." arXiv preprint arXiv:2501.13734 (2025).

2. the paper goes beyond the $f\equiv g$ setting and handles the more realistic validation-based tuning case, which is a genuine extension rather than a cosmetic reformulation. That aligns with the practical hyperparameter tuning problem.

**Weaknesses**:
1. My main weakness is that the paper has no experimental or numerical validation. The whole paper is theoretical, and while that is acceptable in principle, it makes it harder to judge whether the resulting sample-complexity bounds are only formally finite or actually informative in realistic regimes. If they can empirically verify their theory in some simple cases, like weighted group lasso mentioned in Section 8, that will strengthen this paper.

2. The showcased applications in Section 8 are somewhat less compelling than the deep-learning-oriented examples. The current applications, weighted group lasso and weighted fused lasso, are mathematically natural but may feel less appealing. Do authors consider  problems like learning rate schedule in [2] or data reweighting [3].\
[2] Logan Engstrom, Andrew Ilyas, Benjamin Chen, Axel Feldmann, William Moses, and Aleksander Mądry. Optimizing ML Training with Metagradient Descent. 2025.\
[3] Xie, Sang Michael, et al. "Doremi: Optimizing data mixtures speeds up language model pretraining." Advances in Neural Information Processing Systems 36 (2023): 69798-69818.

---

> ### Author Rebuttal · Authors · 2026-03-31
>
> We thank the reviewer for the positive feedback. We are glad that the reviewer found our results to be a meaningful theoretical advance and a realistic extension. We will address the concerns of the reviewer as follows.
>
> ### Questions
> 1. Q: Applicability to modern settings.
>
>     A. We note that our theoretical results can be extended to other modern/deep learning related settings, since the piecewise polynomials/rationals structure that we considered is ubiquitous (also applicable to DNNs, see e.g., [2, 3, 4]), as noted in lines 81-86 in our draft. Concretely, a prior work (which is a strict sub-case of our framework) demonstrates many deep-learning-related applications (i.e., tuning graph neural network, etc.).
>
>      To relate to the settings suggested by the reviewer specifically, there is a recent work [1] that provides guarantees for learning the step-size, learning rate schedule, etc., also leveraging the piecewise structures. For example, for tuning learning rate $\eta$, the update will be $w_{t + 1} = w_t - \eta \nabla f(w_t)$, and if the function $f$ admits piecewise structure, which is the case for DNNs (see e.g., [2,3,4]), $\nabla f(w)$ also admits piecewise structure, and we can apply our results to this case (with the number of quantifier alternations $K = 0$). However, we note that our result is not application-specific and much more general: it can be applicable to: (1) tuning involved in bi-level or multi-level structure ($K \geq 1$), (2) tuning with validation loss ($f \not \equiv g$), and (3) tuning with the loss beyond piecewise polynomials. For references, the example of tuning the learning rate above corresponds to $K = 0$, $f \equiv g$, and the loss $f(w)$ admits a piecewise polynomial structure, __which is a basic case in our framework.__
>
>       However, since our original goal is to establish a general result for the piecewise polynomials/rationals settings, and extending to the more general semi-algebraic structure, which is an open question posed by prior works, we put less emphasis on the applicability. Moreover, we note that the applications in our works are also novel contributions, and we were not be able to link to other application-specific results in prior works due to the page limit. But **we agree with the reviewer that an elaboration of the applicability would make the paper more compelling and comprehensive**, and **we would spend an extra page to elaborate on this part should the paper be accepted**.
>
> ### Conclusion
>
> We hope that our answers resolve the reviewer’s concerns, and we are happy to address other concerns if needed. Many thanks!
>
> ### References
> [1] Dravyansh Sharma, Gradient Descent with Provably Tuned Learning-rate Schedules, AISTATS’26
>
> [2] Bartlett et al., Almost linear VC dimension bounds for piecewise polynomial networks, NeurIPS’18
>
> [3] Montúfar et al., On the number of linear regions of deep neural networks, NeurIPS’14
>
> [4] Bartlett et al., Nearly-tight VC-dimension and pseudodimension bounds for piecewise linear neural networks, JMLR’19
>
> [5] Bacan et al. Sample complexity of data-driven tuning of model hyperparameters in neural networks with structured parameter-dependent dual function, NeurIPS’25

---

> > ### Author Rebuttal · Reviewer_rbQS · 2026-04-02
> >
> > I thank the authors for their rebuttal and agree that the primary focus of this paper is theoretical. I maintain my positive rating based on the current version of the paper.

---

### Decision · Program_Chairs · 2026-04-30

**Decision:**

Accept (regular)

**Comment:**

The paper proposes a framework for generalization guarantees in data-driven multi-dimensional hyperparameter tuning.
 Using tools from real algebraic geometry and statistical learning theory, it introduces complexity bounds for
 tuning hyperparameters based on validation loss, even in bilevel optimization settings.

 All reviewers agree that the paper propose some strong contributions and that the rebuttal have clarified
 most of their concerns. As such, they are all supporting acceptance.